# Eigen Analysis of Conjugate Kernel and Neural Tangent Kernel

Xiangchao Li [* 1]   Xiao Han [* 2]   Qing Yang [2]

## Abstract

In this paper, we investigate deep feedforward neural networks with random weights. The input data matrix $X$ is drawn from a Gaussian mixture model. We demonstrate that certain eigenvalues of the conjugate kernel and neural tangent kernel may lie outside the support of their limiting spectral measures in the high-dimensional regime. The existence and asymptotic positions of such isolated eigenvalues are rigorously analyzed. Furthermore, we provide a precise characterization of the entrywise limit of the projection matrix onto the eigenspace associated with these isolated eigenvalues. Our findings reveal that the eigenspace captures inherent group features present in $X$. This study offers a quantitative analysis of how group features from the input data evolve through hidden layers in randomly weighted neural networks.

## 1. Introduction

In the last decade, deep neural networks (DNNs) have demonstrated dominating performance in various machine-learning tasks such as computer vision (Krizhevsky et al., 2012; He et al., 2016), natural language processing (Amodei et al., 2016; Brown, 2020; Graves et al., 2013; Torfi et al., 2020) and game playing (Silver et al., 2016). One of the empirical findings is their ability to adapt to the features present in the training data, which is considered a fundamental reason for their superior performance (see, for instance, (Bengio et al., 2013; Donahue et al., 2016; Moniri et al., 2023)). Nowadays, many theoretical results have been established to understand the training and generalization of neural networks. In particular, significant research efforts (Abbe et al., 2022; Ba et al., 2022; Bai & Lee, 2019; Damian et al., 2022; Dandi et al., 2024; Mousavi-Hosseini et al., 2022;

Radhakrishnan et al., 2022; Shi et al., 2022) have demonstrated the advantages of feature learning in neural networks. However, understanding the mechanism of DNNs remains challenging and the current theoretical framework is incomplete, largely due to the heavy over-parameterization and high non-convexity of modern neural networks (Allen-Zhu et al., 2019).

Among the theoretical analyses, the Neural Tangent Kernel (NTK, Jacot et al., 2018) has become a powerful tool for understanding neural networks. It is well established that NTK, which is associated with the nonlinear feature map (Bietti & Mairal, 2019), is able to describe the dynamics of heavily over-parameterized neural networks under a specific initialization (Jacot et al., 2018; Du et al., 2019). It also provides insights into the convergence and generalization properties of very wide DNNs (Xie et al., 2017). The NTK theory has also been extended to other network architectures, such as deep residual networks (Belfer et al., 2024) and deep attention networks (Hron et al., 2020).

The eigenvalues and eigenvectors of both the NTK and its closely related counterpart, the Conjugate Kernel (CK), are crucial in understanding the training behavior and generalization performance of the underlying DNN (Fan & Wang, 2020; Yang & Salman, 2019; Yang, 2020; Engel et al., 2023). A recent line of work has analyzed the spectral properties of such kernel matrices. For instance, one may refer to Pennington & Worah (2017); Fan & Wang (2020); Wang et al. (2023); Wang & Zhu (2024); Belfer et al. (2024); Wang et al. (2024); Murray et al. (2022). The spectral properties of these kernel matrices, particularly spiked or extreme eigenvalues, are often closely associated with certain properties of neural networks, including their memorization capacity (Nguyen et al., 2021), training and generalization performance (Liao et al., 2020), and their ability to capture the low-dimensional signal structure inherent in the learning problem (Benigni & Péché, 2022; Wang et al., 2024; Ba et al., 2023). We are interested in investigating the limiting properties of the isolated eigenvalues and their corresponding eigenvectors of kernel matrices in the high-dimensional regime. Through this research, we aim to gain deeper insights into how features of the input data propagate through DNNs.

In this paper, we study the CK and NTK matrices of deep fully-connected neural networks. Recent studies have

---

[*]Equal contribution  [1]School of Management, University of Science and Technology of China [2]International Institute of Finance, School of Management, University of Science and Technology of China. Correspondence to: Qing Yang <yangq@ustc.edu.cn>.

*Proceedings of the $42^{st}$ International Conference on Machine Learning*, Vancouver, Canada. PMLR 267, 2025. Copyright 2025 by the author(s).

demonstrated that, as the network width approaches infinity, the empirical CK and NTK converge to their expected values (Arora et al., 2019; Jacot et al., 2018). Therefore, we focus exclusively on the expected forms of these kernel matrices throughout this paper. We assume that the input data $\boldsymbol{X} \in \mathbb{R}^{p \times n}$ are independently drawn from a $K$-class Gaussian mixture model (GMM) and are independent of the random weights. The analysis of class-structured data is common in the literature. For example, Papyan et al. (2020) demonstrated that when the training error reaches zero, the last-layer classifiers collapse to the class-means; Liao & Couillet (2018) conducted a spectral analysis of the Gram matrix associated with random feature mappings, while Couillet et al. (2018) analyzed the asymptotic performance of several classical classification methods under GMM assumptions; Ali et al. (2022) investigated the spectral behavior of the CK in a high-dimensional regime; and Gu et al. (2024) derived the deterministic equivalents for both the CK and NTK. Based on the results established by Gu et al. (2024), we further show that a finite number of eigenvalues of these kernel matrices lie outside the support of the limiting measure. Our analysis reveals a possible phase transition (Baik et al., 2005) that depends on the choice of activation functions and the differences in covariance among different classes. We also precisely determine the positions of these isolated eigenvalues and eigenvector alignments.

In this article, by demonstrating that the eigenvectors associated with the isolated eigenvalues may contain information relevant to unsupervised classification (clustering), we establish a connection between the features of the input data and neural networks. The techniques employed in this paper are grounded in random matrix theory, and from a theoretical perspective, our analysis falls within the framework of finite-rank deformation models in this field.

**Notations**: Throughout the paper, we use $\|\cdot\|$ to denote the Euclidean norm for vectors and the spectral norm for matrices. The spectrum of a matrix $\boldsymbol{A}$ is denoted by $\mathrm{Spec}(\boldsymbol{A})$. We denote an all-ones vector of dimension $p$ by $\boldsymbol{1}_p$ and the identity matrix of size $p \times p$ by $\boldsymbol{I}_p$. For two sequences of random matrices $\boldsymbol{A} = \{\boldsymbol{A}_n\}_{n \geq 1}$ and $\boldsymbol{B} = \{\boldsymbol{B}_n\}_{n \geq 1}$, we denote $\boldsymbol{A} \sim \boldsymbol{B}$ if

$$\frac{1}{n}\mathrm{tr}\boldsymbol{D}_n(\boldsymbol{A}_n - \boldsymbol{B}_n) \to 0 \text{ and } \boldsymbol{u}_1^{\mathsf{T}}(\boldsymbol{A}_n - \boldsymbol{B}_n)\boldsymbol{u}_2 \to 0 \ a.s.$$

for all deterministic sequences $\{\boldsymbol{D}_n\}_{n \geq 1}$ and all deterministic vectors $\boldsymbol{u}_i \in \mathbb{R}^n, i = 1, 2$ with bounded norms. The symbols $O(\cdot)$ and $o(\cdot)$ stand for the standard big-O and little-o notations. Moreover, if $\|\boldsymbol{A} - \boldsymbol{B}\| \to 0$ almost surely, we write $\boldsymbol{A} = \boldsymbol{B} + o_{a.s.}(1)$. The Hadamard product between two matrices of the same size is denoted by $\circ$. The distance between two sets $A, B \subset \mathbb{C}$ is denoted by $\mathrm{dist}(A, B)$. Let $a \otimes \nu \oplus b$ denote the law of $ax + b$, where $x \sim \nu$ is a

random variable (or random vector) following distribution $\nu$. The indicator function is represented by $\delta$. For a function $f$, we denote its $i$-th derivative by $f^{(i)}$. Specifically, for $i = 1, 2$, we also use $f' = f^{(1)}$ and $f'' = f^{(2)}$. Additionally, $\|f\| = \sup_x |f(x)|$ denotes the supremum norm of $f$. The notation 0 may indicate a zero value, a zero vector or a zero matrix in this paper, changing from line to line. We use $c$ and $C$ to denote positive constants, whose values may change from one line to the next.

## 2. Preliminaries

Let $\boldsymbol{x}_1, ..., \boldsymbol{x}_n \in \mathbb{R}^p$ be $n$ random vectors independently drawn from one of the $K$-class Gaussian mixtures $\mathcal{C}_1, ..., \mathcal{C}_K$, that is

$$\boldsymbol{x}_i \in \mathcal{C}_a \iff \boldsymbol{x}_i \sim \mathcal{N}(0, \ p^{-1}\boldsymbol{C}_a)$$

for some non-negative definite matrix $\boldsymbol{C}_a$. For each $a \in \{1, ..., K\}$, class $\mathcal{C}_a$ has cardinality $n_a$, satisfying $n_1 + \cdots n_K = n$. Write the input features in a matrix $\boldsymbol{X} = [\boldsymbol{x}_1, ..., \boldsymbol{x}_n] \in \mathbb{R}^{p \times n}$. We define the fully-connected DNN with $L$ hidden layers by

$$\boldsymbol{X}_\ell = \frac{1}{\sqrt{d_\ell}}\sigma_\ell(\boldsymbol{W}_\ell \boldsymbol{X}_{\ell-1}) \in \mathbb{R}^{d_\ell \times n} \quad \text{for} \quad \ell = 1, ..., L,$$

with weight matrices $\boldsymbol{W}_\ell \in \mathbb{R}^{d_\ell \times d_{\ell-1}}$ (with convention $d_0 = p$, $\boldsymbol{X}_0 = \boldsymbol{X}$) and nonlinear activation functions $\sigma_1, ..., \sigma_L$ applied entrywise. The CK of the $\ell$-th layer is given by the Gram matrix

$$\boldsymbol{K}_{\mathrm{CK},\ell} = \boldsymbol{K}_{\mathrm{CK},\ell}(\boldsymbol{X}) := \mathbb{E}[\boldsymbol{X}_\ell^{\mathsf{T}}\boldsymbol{X}_\ell] \in \mathbb{R}^{n \times n},$$

where the expectation is taken with respect to the random weights $\boldsymbol{W}_1, ..., \boldsymbol{W}_L$. Following Bietti & Mairal (2019) and Jacot et al. (2018), the CK satisfies that

$$[\boldsymbol{K}_{\mathrm{CK},\ell}]_{ij} = \mathbb{E}[\sigma_\ell(u)\sigma_\ell(v)] \tag{1}$$

with

$$[u, v]^{\mathsf{T}} \sim \mathcal{N}\left(\boldsymbol{0}, \begin{bmatrix} [\boldsymbol{K}_{\mathrm{CK},\ell-1}]_{ii} & [\boldsymbol{K}_{\mathrm{CK},\ell-1}]_{ij} \\ [\boldsymbol{K}_{\mathrm{CK},\ell-1}]_{ji} & [\boldsymbol{K}_{\mathrm{CK},\ell-1}]_{jj} \end{bmatrix}\right),$$

while the NTK denoted by $\boldsymbol{K}_{\mathrm{NTK},\ell}$ takes the form

$$\begin{aligned} \boldsymbol{K}_{\mathrm{NTK},\ell} &= \boldsymbol{K}_{\mathrm{CK},\ell} + \boldsymbol{K}_{\mathrm{NTK},\ell-1} \circ \boldsymbol{K}'_{\mathrm{CK},\ell}, \\ \boldsymbol{K}_{\mathrm{NTK},0} &= \boldsymbol{K}_{\mathrm{CK},0} = \boldsymbol{X}^{\mathsf{T}}\boldsymbol{X}, \end{aligned} \tag{2}$$

where

$$[\boldsymbol{K}'_{\mathrm{CK},\ell}]_{ij} = \mathbb{E}[\sigma'_\ell(u)\sigma'_\ell(v)].$$

We consider the high-dimensional regime where $n$ and $p$ are comparable and assume the following conditions on the input data, weights and activation functions.

**Assumption 2.1.** As $n \to \infty$, the following conditions hold:

- The ratios $\frac{p}{n} \to c_0 \in (0, \infty)$, and $\frac{n_a}{n} \to c_a \in (0, 1)$ for each $a \in \{1, ..., K\}$.

- Denoting $\boldsymbol{C}^\circ = \frac{1}{p} \sum_{a=1}^{K} \frac{n_a}{n} \boldsymbol{C}_a$, $\boldsymbol{C}_a^\circ = \boldsymbol{C}_a - \boldsymbol{C}^\circ$ and $\tau_0 = \sqrt{\operatorname{tr}\boldsymbol{C}^\circ/p}$, we have $\lim_{n\to\infty} \tau_0 \in (0, \infty)$.

- For $a, b \in \{1, ..., K\}$, it holds that $\|\boldsymbol{C}_a\| = O(1)$ and $\operatorname{tr}\boldsymbol{C}_a\boldsymbol{C}_b = O(p)$.

This assumption ensures the classification task is feasible and non-trivial (see Gu et al. (2024) for example).

**Assumption 2.2.** The random weights $\boldsymbol{W}_1 \in \mathbb{R}^{d_1 \times p}$,..., $\boldsymbol{W}_L \in \mathbb{R}^{d_L \times d_{L-1}}$ are independent. Moreover, the entries of $\boldsymbol{W}_\ell$ are i.i.d. and satisfy

$$\mathbb{E}[\boldsymbol{W}_\ell]_{ij} = 0, \quad \mathbb{E}[\boldsymbol{W}_\ell]_{ij}^2 = 1, \quad \mathbb{E}[\boldsymbol{W}_\ell]_{ij}^4 < \infty.$$

This assumption on the random weights is relatively mild and holds for many common weights such as i.i.d. standard Gaussian weights.

**Assumption 2.3.** Let $\xi \sim \mathcal{N}(0, 1)$. The activation functions $\sigma_1, ..., \sigma_L$ satisfy that

$$\max_{k \in \{0,1,2,3,4\}} \mathbb{E}[\sigma_\ell^{(k)}(\xi)] < C, \quad 1 \le \ell \le L \qquad (3)$$

for some universal constant $C$.

The boundness of $\mathbb{E}[\sigma_\ell^{(k)}(\xi)]$ in (3) is needed to derive the deterministic equivalents for the CK and NTK. Moreover, as noted by Gu et al. (2024), for non-differentiable functions, Assumption 2.3 holds if $|\sigma_\ell|$ is bounded above by some polynomial function through the application of Gaussian integration by parts: $\mathbb{E}\sigma'(\xi) = \mathbb{E}\xi\sigma(\xi)$. Consequently, Assumption 2.3 accommodates commonly used activation functions such as ReLU, Sigmoid, and Tanh, provided that these activation functions are centered and normalized to ensure compliance with (3).

Before presenting the preliminary theoretical results, it is necessary to introduce several key expressions:

$$\boldsymbol{\psi} = \{\psi_i\}_{i=1}^n := \{\|\boldsymbol{x}_i\|^2 - \mathbb{E}\|\boldsymbol{x}_i\|^2\}_{i=1}^n \in \mathbb{R}^n,$$

$$\boldsymbol{t} := \left\{ \frac{1}{\sqrt{p}} \operatorname{tr}\boldsymbol{C}_a^o \right\}_{a=1}^K \in \mathbb{R}^K, \qquad (4)$$

$$\boldsymbol{T} := \left\{ \frac{1}{p} \operatorname{tr} \boldsymbol{C}_a\boldsymbol{C}_b \right\}_{a,b=1}^K \in \mathbb{R}^{K \times K}.$$

The following two lemmas, proved by Gu et al. (2024), provide asymptotic equivalents for the CK and NTK.

**Lemma 2.4.** *(Asymptotic spectral equivalent for the CK (Gu et al., 2024)). Suppose Assumptions 2.1-2.3 hold. Let $\tau_0, \tau_1..., \tau_L$ be a sequence of non-negative numbers satisfying the recursion:*

$$\tau_\ell = \sqrt{\mathbb{E}[\sigma_\ell^2(\tau_{\ell-1}\xi)]}, \quad \ell \in \{1, ..., L\}.$$

*We further assume $\mathbb{E}[\sigma_\ell(\tau_{\ell-1}\xi)] = 0$. Then as $n \to \infty$, we have*

$$\|\boldsymbol{K}_{\mathrm{CK},\ell}(\boldsymbol{X}) - \widetilde{\boldsymbol{K}}_{\mathrm{CK},\ell}(\boldsymbol{X})\| \to 0 \ \ a.s.,$$
$$\widetilde{\boldsymbol{K}}_{\mathrm{CK},\ell} := \alpha_{\ell,1}\boldsymbol{X}^\mathsf{T}\boldsymbol{X} + \boldsymbol{V}\boldsymbol{A}_\ell\boldsymbol{V}^\mathsf{T} + \alpha_{\ell,0}\boldsymbol{I}_n, \qquad (5)$$

*where $\alpha_{\ell,0} = \tau_\ell^2 - \tau_0^2\alpha_{\ell,1} \ge 0$,*

$$\boldsymbol{V} := [\boldsymbol{J}/\sqrt{p}, \boldsymbol{\psi}] \in \mathbb{R}^{n \times (K+1)},$$
$$\boldsymbol{A}_\ell := \begin{bmatrix} \alpha_{\ell,2}\boldsymbol{t}\boldsymbol{t}^\mathsf{T} + \alpha_{\ell,3}\boldsymbol{T} & \alpha_{\ell,2}\boldsymbol{t} \\ \alpha_{\ell,2}\boldsymbol{t}^\mathsf{T} & \alpha_{\ell,2} \end{bmatrix} \in \mathbb{R}^{(K+1)\times(K+1)}, \qquad (6)$$

*for class label vectors $\boldsymbol{J} = [\boldsymbol{j}_1, ..., \boldsymbol{j}_K] \in \mathbb{R}^{n \times K}$ with $\boldsymbol{j}_a = \{\delta_{x_i \in \mathcal{C}_a}\}_{i=1}^n$. The non-negative scalars $\alpha_{\ell,1}, \alpha_{\ell,2}, \alpha_{\ell,3}$ satisfy the following recursions:*

$$\alpha_{\ell,1} = \mathbb{E}[\sigma_\ell'(\tau_{\ell-1}\xi)]^2 \alpha_{\ell-1,1},$$
$$\alpha_{\ell,2} = \mathbb{E}[\sigma_\ell'(\tau_{\ell-1}\xi)]^2 \alpha_{\ell-1,2} + \frac{1}{4}\mathbb{E}[\sigma_\ell''(\tau_{\ell-1}\xi)]^2 \alpha_{\ell-1,4}^2,$$
$$\alpha_{\ell,3} = \mathbb{E}[\sigma_\ell'(\tau_{\ell-1}\xi)]^2 \alpha_{\ell-1,3} + \frac{1}{2}\mathbb{E}[\sigma_\ell''(\tau_{\ell-1}\xi)]^2 \alpha_{\ell-1,1}^2$$
$$\qquad (7)$$

*with*

$$\alpha_{\ell,4} = \alpha_{\ell-1,4}\mathbb{E}[(\sigma_\ell'(\tau_{\ell-1}\xi))^2 + \sigma_\ell(\tau_{\ell-1}\xi)\sigma_\ell''(\tau_{\ell-1}\xi)],$$
$$\alpha_{0,1} = \alpha_{0,4} = 1, \ \alpha_{0,2} = \alpha_{0,3} = 0.$$

**Lemma 2.5.** *(Asymptotic spectral equivalent for the NTK (Gu et al., 2024)). Suppose Assumptions 2.1-2.3 hold under the same notation and settings as in Lemma 2.4. Let $\dot{\tau}_0 = 0$, $\dot{\tau}_1, ..., \dot{\tau}_L \ge 0$ be a sequence non-negative numbers such that*

$$\dot{\tau}_\ell = \sqrt{\mathbb{E}[\sigma_\ell'(\tau_{\ell-1}\xi)]^2}, \quad \ell \in \{1, ..., L\}$$

*and let $\kappa_\ell^2 = \tau_\ell^2 + \dot{\tau}_\ell^2$ with $\kappa_0 = \tau_0$. Then as $n \to \infty$, we have*

$$\|\boldsymbol{K}_{\mathrm{NTK},\ell}(\boldsymbol{X}) - \widetilde{\boldsymbol{K}}_{\mathrm{NTK},\ell}(\boldsymbol{X})\| \to 0 \ \ a.s.,$$
$$\widetilde{\boldsymbol{K}}_{\mathrm{NTK},\ell} := \beta_{\ell,1}\boldsymbol{X}^\mathsf{T}\boldsymbol{X} + \boldsymbol{V}\boldsymbol{B}_\ell\boldsymbol{V}^\mathsf{T} + \beta_{\ell,0}\boldsymbol{I}_n, \qquad (8)$$

*where $\beta_{\ell,0} = \kappa_\ell^2 - \tau_0^2\beta_{\ell,1} \ge 0$ and*

$$\boldsymbol{B}_\ell := \begin{bmatrix} \beta_{\ell,2}\boldsymbol{t}\boldsymbol{t}^\mathsf{T} + \beta_{\ell,3}\boldsymbol{T} & \beta_{\ell,2}\boldsymbol{t} \\ \beta_{\ell,2}\boldsymbol{t}^\mathsf{T} & \beta_{\ell,2} \end{bmatrix} \in \mathbb{R}^{(K+1)\times(K+1)}.$$

*The non-negative scalars $\beta_{\ell,1}, \beta_{\ell,2}, \beta_{\ell,3}$ satisfy*

$$\beta_{\ell,1} = \alpha_{\ell,1} + \mathbb{E}\left[\sigma_\ell'(\tau_{l-1}\xi)\right]^2 \beta_{\ell-1,1},$$
$$\beta_{\ell,2} = \alpha_{\ell,2} + \mathbb{E}\left[\sigma_\ell'(\tau_{\ell-1}\xi)\right]^2 \beta_{\ell-1,2},$$
$$\beta_{\ell,3} = \alpha_{\ell,3} + \mathbb{E}\left[\sigma_\ell'(\tau_{\ell-1}\xi)\right]^2 \beta_{\ell-1,3}$$
$$\qquad + \mathbb{E}\left[\sigma_\ell''(\tau_{\ell-1}\xi)\right]^2 \alpha_{\ell-1,1}\beta_{\ell-1,1} \qquad (9)$$

*with $\beta_{0,1} = 1, \beta_{0,2} = \beta_{0,3} = 0$.*

For an $p \times p$ matrix $\boldsymbol{M}$ with real eigenvalues $\lambda_1^{\boldsymbol{M}} \geq \cdots \lambda_p^{\boldsymbol{M}}$, its empirical spectral distribution is defined as

$$F^{\boldsymbol{M}}(x) = \frac{1}{p} \#\{i = 1, ..., p, \ \lambda_i^{\boldsymbol{M}} \leq x\},$$

and the probability measure induced by $F^{\boldsymbol{M}}$ is denoted by $\mu_{\boldsymbol{M}}$. We define the Stieltjes transform of a probability measure $\nu$ as

$$m_\nu(z) = \int \frac{1}{\lambda - z} \mathrm{d}\nu(\lambda), \ z \in \mathbb{C}^+ = \{z \colon z \in \mathbb{C}, \ \Im z > 0\}.$$

It admits a natural extension to the lower-half of the complex space by the fact that

$$m_G(z) = \overline{m}_G(\bar{z}), \ z \in \mathbb{C}^- = \{z : z \in \mathbb{C}, \ \Im z < 0\}.$$

Prior to presenting our results, we need the following lemma, which guarantees the existence of the limiting spectral distribution of $\boldsymbol{X}^\top \boldsymbol{X}$. This lemma is established by Benaych-Georges & Couillet (2016).

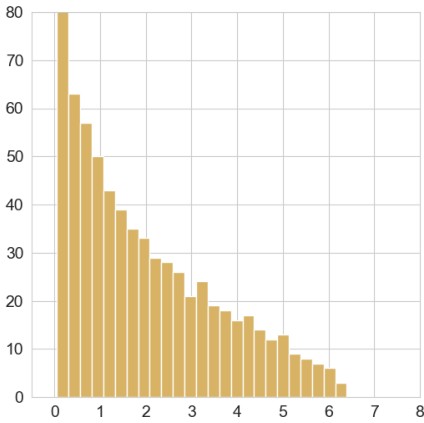

*Figure 1.* Spectrum of $\widetilde{\boldsymbol{K}}_{\mathrm{CK},3}$ under the parameter settings: $n = 1200$, $p = 600$, $c_1 = c_3 = 0.3$, $c_2 = 0.4$ and $\boldsymbol{C}_a = (1 + 2(a-1)/\sqrt{p})\boldsymbol{I}_p$ for $a = 1, 2, 3$. The activation functions for the three layers are $\sigma_1 = \sigma_2 = \sigma_3 = \mathrm{Poly}$, where $\mathrm{Poly}(t) = 0.2t^2 + t$. The weights $\boldsymbol{W}_1 \in \mathbb{R}^{d_1 \times p}$, $\boldsymbol{W}_2 \in \mathbb{R}^{d_2 \times d_1}$ and $\boldsymbol{W}_3 \in \mathbb{R}^{d_3 \times d_2}$ consist of i.i.d. standard normal entries, where $d_1 = d_2 = 2000$, $d_3 = 1000$.

**Lemma 2.6.** *Suppose Assumption 2.1 holds. For $z \in \mathbb{C}$, we define the resolvents*

$$\boldsymbol{G}(z) = (\boldsymbol{X}^\top \boldsymbol{X} - z\boldsymbol{I}_n)^{-1}, \quad \underline{\boldsymbol{G}}(z) = (\boldsymbol{X}\boldsymbol{X}^\top - z\boldsymbol{I}_p)^{-1}.$$

*Then as $n \to \infty$, we have*

$$\boldsymbol{G}(z) \sim \boldsymbol{Q}_1(z) := c_0 \operatorname{diag}\{m_a(z)1_{n_a}\}_{a=1}^K \qquad (10)$$

*and*

$$\underline{\boldsymbol{G}}(z) \sim \boldsymbol{Q}_2(z) := -\frac{1}{z}\left(\boldsymbol{I}_p + \sum_{a=1}^K c_a m_a(z)\boldsymbol{C}_a\right)^{-1}, \qquad (11)$$

*where $\{m_a(z)\}_{a=1}^K \in \mathbb{C}^K$ is a unique vector such that*

$$\Im z \Im m_a(z) \geq 0, \ \Im z \Im(z m_a(z)) \geq 0, \ c_0|m_a(z)| \leq (\Im z)^{-1}$$

*and*

$$c_0 m_a(z) = -\frac{1}{z}\frac{1}{1 + \widetilde{m}_a(z)},$$

$$\widetilde{m}_a(z) = -\frac{1}{z}\frac{1}{p} \operatorname{tr} \boldsymbol{C}_a \left(\boldsymbol{I}_p + \sum_{b=1}^k c_b m_b(z)\boldsymbol{C}_b\right)^{-1} \qquad (12)$$

$$= \frac{1}{p}\operatorname{tr}\boldsymbol{C}_a\boldsymbol{Q}_2(z).$$

*Besides, $c_0 m_1(z), ..., c_0 m_K(z)$ are Stieltjes transforms of some $\mathbb{R}^+$-compactly supported probability measures $\nu_1, ..., \nu_K$. The probability measure $\mu$ defined by the Stieltjes transform*

$$m_\mu(z) = c_0 \sum_{a=1}^K c_a m_a(z) \qquad (13)$$

*is a deterministic probability measure with compact support $\mathcal{S} = \bigcup_{a=1}^K \operatorname{supp}(\nu_a)$ such that*

$$\mu_{\boldsymbol{X}^\top\boldsymbol{X}} \to \mu \text{ in distribution,} \qquad (14)$$

$$\operatorname{dist}(\operatorname{Spec}(\boldsymbol{X}^\top\boldsymbol{X}), \ \mathcal{S} \cup \{0\}) \to 0 \qquad (15)$$

*almost surely. For $x \in \mathbb{R}$ such that $\operatorname{dist}(x, \mathcal{S} \cup \{0\}) > c$ for some positive constant $c$, (12) also holds.*

According to Lemma 2.4 and Lemma 2.5, the $\ell$-th layer of kernel matrices can be approximated by a linear combination of $\boldsymbol{X}^\top \boldsymbol{X}$, $\boldsymbol{I}_n$ and a low rank matrix. Based on above results and Lemma A.4, it follows immediately that $\|F^{\boldsymbol{K}_{\mathrm{CK},\ell}} - F^{\widetilde{\boldsymbol{K}}_{\mathrm{CK},\ell}}\| \to 0$ ($\|F^{\boldsymbol{K}_{\mathrm{NTK},\ell}} - F^{\widetilde{\boldsymbol{K}}_{\mathrm{NTK},\ell}}\| \to 0$). Therefore, the LSD of $\boldsymbol{K}_{\mathrm{CK},\ell}(\boldsymbol{K}_{\mathrm{NTK},\ell})$ can be written as $\mu_{\mathrm{CK},\ell} = \alpha_{\ell,1} \otimes \mu \oplus \alpha_{\ell,0}$ ($\mu_{\mathrm{NTK},\ell} = \beta_{\ell,1} \otimes \mu \oplus \beta_{\ell,0}$).

To visualize the ESD of the CK, we present the empirical spectral distribution of $\widetilde{\boldsymbol{K}}_{\mathrm{CK},3}$ in Figure 1, obtained using the same polynomial activation function across all layers. As shown, this ESD does not exhibit any isolated eigenvalues. In the following section, we will investigate scenarios in which eigenvalues detach from the bulk.

## 3. Main results: eigen analysis of the CK and NTK

In this section, we follow the notation established in Lemmas 2.4-2.5, and without loss of generality, assume that

$$\boldsymbol{x}_i \in \mathcal{C}_a \quad \text{for} \quad 1 + \sum_{j=1}^{a-1} n_j \leq i \leq \sum_{j=1}^a n_j.$$

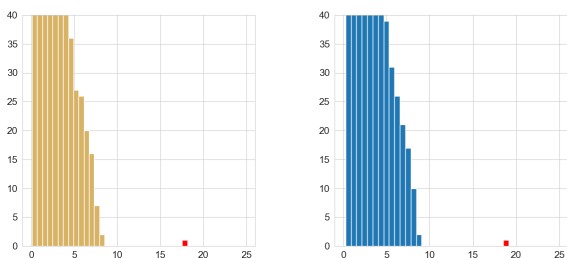

(a) Eigenvalues histogram of $\widetilde{\boldsymbol{K}}_{\mathrm{CK},3}$  (b) Eigenvalues histogram of $\boldsymbol{K}_{\mathrm{CK},3}$

*Figure 2.* Spectra of $\widetilde{\boldsymbol{K}}_{\mathrm{CK},3}$ and $\boldsymbol{K}_{\mathrm{CK},3}$ (with the expectation estimated from 500 realizations of weights).

This assumption is also adopted in the simulations. We denote the supports of the limiting spectral measures of $\boldsymbol{K}_{\mathrm{CK},\ell}$ and $\boldsymbol{K}_{\mathrm{NTK},\ell}$ by $\mathcal{S}_{\mathrm{CK},\ell}$ and $\mathcal{S}_{\mathrm{NTK},\ell}$, respectively. According to Lemma 2.4, Lemma 2.5 and Lemma 2.6, we have

$$\mathcal{S}_{\mathrm{CK},\ell} = \alpha_{\ell,1}\mathcal{S} + \alpha_{\ell,0}, \quad \mathcal{S}_{\mathrm{NTK},\ell} = \beta_{\ell,1}\mathcal{S} + \beta_{\ell,0}.$$

### 3.1. Isolated eigenvalues

This subsection provides a precise characterization of the isolated eigenvalues of the kernel matrices. Based on the findings from the previous section, it suffices to focus on the isolated eigenvalues of $\widetilde{\boldsymbol{K}}_{\mathrm{CK},\ell}$ and $\widetilde{\boldsymbol{K}}_{\mathrm{NTK},\ell}$. To illustrate this, we visualize the spectra of $\widetilde{\boldsymbol{K}}_{\mathrm{CK},\ell}$ and $\boldsymbol{K}_{\mathrm{CK},\ell}$ in Figure 2, for the case where $\boldsymbol{C}_a = (1 + 8(a-1)/\sqrt{p})\boldsymbol{I}_p$, with all other settings kept consistent with those in Figure 1.

We now present the results for the eigenvalues of the CK and NTK that detach from the bulk of their spectrum, focusing on the non-trivial case where $\alpha_{\ell,1}$ ($\beta_{\ell,1}$) is bounded away from zero. The following theorem establishes the asymptotic behavior of the isolated eigenvalues of $\boldsymbol{K}_{\mathrm{CK},\ell}$.

**Theorem 3.1.** *(Isolated eigenvalues of the CK). Define the function $h_\ell^1(z) = 1 + \frac{\alpha_{\ell,2}}{\alpha_{\ell,1}}\frac{1}{p}\sum_{a=1}^K c_a m_a(z)\mathrm{tr}\boldsymbol{C}_a^2$ and the set $\mathcal{H}_{1,\ell}^p = \{z \in \mathbb{C} | h_\ell^1(z) = 0\}$. Denote*

$$\mathcal{S}_{1,\ell} = \mathcal{S} \cup \mathcal{H}_{1,\ell}^p \cup \{0\},$$

*and*

$$\underline{\boldsymbol{H}}_1(z) = \left[ h_\ell^1(z)\boldsymbol{I}_K + \frac{\alpha_{\ell,3}}{\alpha_{\ell,1}}h_\ell^1(z)\boldsymbol{T}\boldsymbol{\Gamma}(z) + \frac{\alpha_{\ell,2}}{\alpha_{\ell,1}}\boldsymbol{t}\boldsymbol{t}^\mathsf{T}\boldsymbol{\Gamma}(z) \right],$$

*where $\boldsymbol{T}$ is defined in (4) and*

$$\boldsymbol{\Gamma}(z) = \mathrm{diag}\{c_a m_a(z)\}_{a=1}^K.$$

*Under Assumptions 2.1-2.3, if $\underline{\boldsymbol{H}}_1(\rho)$ has a zero eigenvalue with multiplicity $k_\rho$ and $\mathrm{dist}(\rho, \mathcal{S}_{1,\ell}) > C$ for some positive*

constant $C$, then $\boldsymbol{K}_{\mathrm{CK},\ell}$ has $k_\rho$ eigenvalues

$$\lambda_j^{CK} \geq \cdots \geq \lambda_{j+k_\rho-1}^{CK}$$

*outside $\mathcal{S}_{\mathrm{CK},\ell}$ such that*

$$\max_{0 \leq i \leq k_\rho-1} |\lambda_{j+i}^{CK} - (\alpha_{\ell,1}\rho + \alpha_{\ell,0})| \to 0 \quad a.s. \quad (16)$$

*Moreover, if there exists a $\rho_+ \in \mathcal{S}^c$ satisfying $\det\boldsymbol{H}_1(\rho_+) = 0$ and $\rho_+ \to \rho \in \mathcal{H}_{1,\ell}^p$, where $\boldsymbol{H}_1(z)$ is defined in (28), then there are $k_\rho^+$ eigenvalues of $\boldsymbol{K}_{\mathrm{CK},\ell}$,*

$$\lambda_j^{CK} \geq \cdots \geq \lambda_{j+k_\rho^+-1}^{CK}$$

*outside $\mathcal{S}_{\mathrm{CK},\ell}$ such that*

$$\max_{0 \leq i \leq k_\rho^+-1} |\lambda_{j+i}^{CK} - (\alpha_{\ell,1}\rho + \alpha_{\ell,0})| \to 0 \quad a.s., \quad (17)$$

*where $k_\rho^+$ is the multiplicity of zero as an eigenvalue of $\boldsymbol{H}_1(\rho_+)$.*

Lemma 2.5 shows that the asymptotic properties of the eigenvalues of the CK matrices also hold for the NTK, up to a change of the associated coefficients $\alpha_{\ell,i}$ to $\beta_{\ell,i}$ for $i = 1, 2, 3, 4$. Therefore, we can immediately obtain the following theorem, which describes the behaviors of the isolated eigenvalues of the NTK.

**Theorem 3.2.** *(Isolated eigenvalues of the NTK). Define $h_\ell^2(z) := 1 + \frac{\beta_{\ell,3}}{\beta_{\ell,1}}\frac{1}{p}\sum_{a=1}^K c_a m_a(z)\mathrm{tr}\boldsymbol{C}_a^2$ and the set $\mathcal{H}_{2,\ell}^p := \{z \in \mathbb{C} | h_\ell^2(z) = 0\}$. Denote $\mathcal{S}_{2,\ell} = \mathcal{S} \cup \mathcal{H}_{2,\ell}^p \cup \{0\}$ and*

$$\underline{\boldsymbol{H}}_2(z) := \left[ h_\ell^2(z)\boldsymbol{I}_K + \frac{\beta_{\ell,3}}{\beta_{\ell,1}}h_\ell^2(z)\boldsymbol{T}\boldsymbol{\Gamma}(z) + \frac{\beta_{\ell,2}}{\beta_{\ell,1}}\boldsymbol{t}\boldsymbol{t}^\mathsf{T}\boldsymbol{\Gamma}(z) \right].$$

*Suppose Assumptions 2.1-2.3 hold, for $\rho$ being a solution to $\det\underline{\boldsymbol{H}}_2(\rho) = 0$ with mulitiplicity $k_\rho$ and $\mathrm{dist}(\rho, \mathcal{S}_{2,\ell}) > C$ for some positive constant $C$, we conclude that there exists $k_\rho$ eigenvalues of $\boldsymbol{K}_{\mathrm{NTK},\ell}$,*

$$\lambda_j^{\mathrm{NTK}} \geq \cdots \geq \lambda_{j+k_\rho-1}^{\mathrm{NTK}}$$

*outside $\mathcal{S}_{\mathrm{NTK},\ell}$ such that*

$$\max_{0 \leq i \leq k_\rho-1} |\lambda_{j+i}^{\mathrm{NTK}} - (\beta_{\ell,1}\rho + \beta_{\ell,0})| \to 0 \quad a.s.,$$

*where the scalars $\beta_{\ell,0}$ and $\beta_{\ell,1}$ are defined in (9) and $k_\rho$ is the multiplicity of zero as an eigenvalue of $\underline{\boldsymbol{H}}_2(\rho)$. We denote $\boldsymbol{H}_2(z)$ by replacing the $\alpha$'s in $\boldsymbol{H}_1(z)$ with $\beta$'s. For $\rho_+ \in \mathcal{S}^c$ being a solution to $\det\boldsymbol{H}_2(z) = 0$ and $\rho_+ \to \rho \in \mathcal{H}_{2,\ell}^p$, there are $k_\rho^+$ eigenvalues of $\boldsymbol{K}_{\mathrm{NTK},\ell}$,*

$$\lambda_j^{\mathrm{NTK}} \geq \cdots \geq \lambda_{j+k_\rho^+-1}^{\mathrm{NTK}}$$

*such that*

$$\max_{0 \leq i \leq k_\rho^+-1} |\lambda_{j+i}^{\mathrm{NTK}} - (\beta_{\ell,1}\rho + \beta_{\ell,0})| \to 0 \quad a.s.,$$

*where $k_\rho^+$ is the multiplicity of zero as an eigenvalue of $\boldsymbol{H}_2(\rho_+)$.*

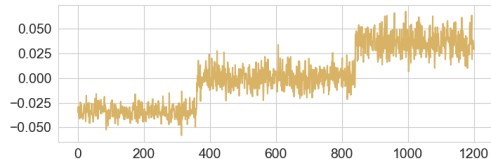

(a) Eigenvector associated with the isolated eigenvalue of $\widetilde{\boldsymbol{K}}_{\mathrm{CK},3}$.

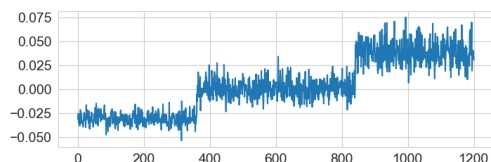

(b) Eigenvector associated with the isolated eigenvalue of $\boldsymbol{K}_{\mathrm{CK},3}$.

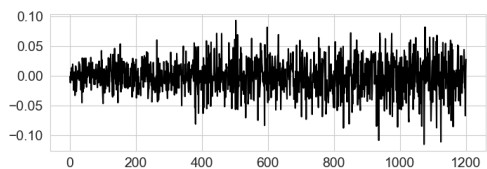

(c) Eigenvector associated with the first eigenvalue of $\boldsymbol{X}^\mathsf{T}\boldsymbol{X}$.

*Figure 3.* Visualization of the eigenvectors corresponding to the first eigenvalues of matrices $\widetilde{\boldsymbol{K}}_{\mathrm{CK},\ell}$, $\boldsymbol{K}_{\mathrm{CK},\ell}$ and $\boldsymbol{X}^\mathsf{T}\boldsymbol{X}$ under the setting of Figure 2. In (a), $\boldsymbol{\omega}_1 = [-0.643, 0.005, 0.693]$ while in (c), $\boldsymbol{\omega}_1 = [-0.008, 0.036, 0.008]$.

Theorem 3.1 and Theorem 3.2 establish the conditions for the occurrence of isolated eigenvalues and provide their asymptotic positions.

### 3.2. Behaviors of the eigenvectors

The preceding subsection studied the eigenvalues of the kernel matrices. We now turn our attention to the asymptotic properties of the eigenvectors associated with the isolated eigenvalues. Specifically, we provide a detailed analysis of the CK matrices, as the NTK matrices exhibit a similar pattern.

We begin with a toy empirical example. Under the same setting as Figure 2, Figures 3(a)-(b) display the eigenvectors associated with the isolated eigenvalues of $\widetilde{\boldsymbol{K}}_{\mathrm{CK},3}$ and $\boldsymbol{K}_{\mathrm{CK},3}$, respectively. Figure 3(c) shows the eigenvector corresponding to the leading eigenvalue of $\boldsymbol{X}^\mathsf{T}\boldsymbol{X}$.

From these figures, we can reasonably infer that the eigenvectors corresponding to the isolated eigenvalues (Figures 3(a)-(b)) are more informative, as they likely capture relevant information for clustering. In contrast, the eigenvector depicted in Figure 3 (c) can be reasonably regarded as pure

noise. Inspired by these observations, an interesting question arises:

**How can we understand the structure of the eigenvector (or eigenspace) associated with an isolated eigenvalue in a high-dimensional regime?**

We use $\mathrm{Span}(\boldsymbol{J})$ to denote the space spanned by $\boldsymbol{j}_a, a = 1, ..., K$. Let $\boldsymbol{P}_{\boldsymbol{J}}$ be the projection onto $\mathrm{Span}(\boldsymbol{J})$. The eigenvector associated with the $i$th isolated eigenvalue $\lambda_i^{CK}$ is denoted by $\hat{\boldsymbol{u}}_i$. This eigenvector can be decomposed into a signal part and a random part:

$$\hat{\boldsymbol{u}}_i = \boldsymbol{P}_{\boldsymbol{J}}\hat{\boldsymbol{u}}_i + (\boldsymbol{I}_n - \boldsymbol{P}_{\boldsymbol{J}})\hat{\boldsymbol{u}}_i = \sum_{a=1}^{K}\left(\omega_{ia}\frac{\boldsymbol{j}_a}{\sqrt{n_a}} + \sigma_a \boldsymbol{v}_a\right),$$
(18)

where $\boldsymbol{v}_a$ is a random unit vector orthogonal to the $\boldsymbol{j}_a$, and its entries are identically distributed. The scalar $\omega_{ia} = \hat{\boldsymbol{u}}_i^\mathsf{T}\frac{\boldsymbol{j}_a}{\sqrt{n_a}}$ measures the cosine between $\hat{\boldsymbol{u}}_i$ and $\frac{\boldsymbol{j}_a}{\sqrt{n_a}}$, and $\sigma_a$ quantifies the extent of fluctuations in $\omega_{ia}\frac{\boldsymbol{j}_a}{\sqrt{n_a}}$. We denote $\boldsymbol{\omega}_i = \{\omega_{ia}\}_{a=1}^{K}$.

If the isolated eigenvalue $\lambda_i^{CK}$ is simple, it follows from $\omega_{ia} = \hat{\boldsymbol{u}}_i^\mathsf{T}\frac{\boldsymbol{j}_a}{\sqrt{n_a}}$ that

$$\frac{1}{n_a}[\boldsymbol{J}^\mathsf{T}\hat{\boldsymbol{u}}_i\hat{\boldsymbol{u}}_i^\mathsf{T}\boldsymbol{J}]_{aa} = \frac{\boldsymbol{j}_a^\mathsf{T}\hat{\boldsymbol{u}}_i\hat{\boldsymbol{u}}_i^\mathsf{T}\boldsymbol{j}_a}{n_a} = \omega_{ia}^2.$$
(19)

Thus, if $\rho$ identified in Theorem 3.1 satisfies that $\underline{\boldsymbol{H}}_1(\rho)$ has a simple (multiplicity one) zero eigenvalue, we can explicitly evaluate $\omega_a$. Moreover, we say an eigenvector $\hat{\boldsymbol{u}}_i$ is non-informative for clustering (i.e., it does not contain any information about $\boldsymbol{J}$) if $\max_a \hat{\boldsymbol{u}}_i^\mathsf{T}\frac{\boldsymbol{j}_a}{\sqrt{n_a}} = o_{a.s.}(1)$, meaning that $\hat{\boldsymbol{u}}_i$ becomes asymptotically orthogonal to $\mathrm{Span}(\boldsymbol{J})$. It is important to note that the eigenvector is not unique, and essentially, our analysis concerns the asymptotic behavior of the matrix $\frac{1}{n_a}\boldsymbol{J}^\mathsf{T}\hat{\boldsymbol{u}}_i\hat{\boldsymbol{u}}_i^\mathsf{T}\boldsymbol{J}$, as described in (19).

Let $\lambda_j, ..., \lambda_{j+k_\rho-1}$ be a group of isolated eigenvalues of $\boldsymbol{K}_{\mathrm{CK}}$ that converge to the same limit $\alpha_{\ell,1}\rho + \alpha_{\ell,0}$, where $\rho$ is identified in Theorem 3.1. As per Lemma 2.4, the eigenspace

$$\mathrm{Span}\{\hat{\boldsymbol{u}}_i \in \mathbb{S}^{n-1}|\ \boldsymbol{K}_{\mathrm{CK},\ell}\hat{\boldsymbol{u}}_i = \lambda_{j+i-1}\hat{\boldsymbol{u}}_i\}$$
(20)

is asymptotically equivalent to the eigenspace associated with eigenvalues of $\boldsymbol{X}^\mathsf{T}\boldsymbol{X} + \alpha_{\ell,1}^{-1}\boldsymbol{V}\boldsymbol{A}_\ell\boldsymbol{V}^\mathsf{T}$ that have a deterministic limit $\rho$. This relationship is also illustrated in Figure 3. In Definition 3.3 below, we present a precise mathematical criterion for determining when an eigenspace is considered informative.

**Definition 3.3. (Informative eigenspace).** We say the eigenspace defined in (20) is informative if there is a non zero matrix $\mathbf{A}(\boldsymbol{J})$ depending on $\boldsymbol{J}$ such that

$$\|\frac{1}{p}\boldsymbol{J}^\mathsf{T}\widehat{\boldsymbol{\Pi}}_\rho\boldsymbol{J} - \mathbf{A}(\boldsymbol{J})\| = o_{a.s.}(1).$$

Otherwise if $\mathbf{A}(\boldsymbol{J}) = 0$, then the eigenspace is non-informative.

Denote by $\widehat{\boldsymbol{\Pi}}_\rho = \sum_{i=1}^{k_\rho} \hat{\boldsymbol{u}}_i \hat{\boldsymbol{u}}_i^\mathsf{T}$ the projection onto the eigenspace defined in (20). Then we can write $\frac{1}{p} \boldsymbol{j}_a^\mathsf{T} \widehat{\boldsymbol{\Pi}}_\rho \boldsymbol{j}_b$ as

$$
-\frac{1}{2\pi\mathrm{i}} \oint_{\partial\gamma_\rho} \frac{1}{p} \boldsymbol{j}_a^\mathsf{T} (\boldsymbol{X}^\mathsf{T}\boldsymbol{X} + \alpha_{\ell,1}^{-1} \boldsymbol{V}\boldsymbol{A}_\ell \boldsymbol{V}^\mathsf{T} - z\boldsymbol{I}_n)^{-1} \boldsymbol{j}_b \mathrm{d}z
$$
$$
+ o_{a.s.}(1),
$$
(21)

where $\gamma_\rho$ is an open disc such that $\rho$ belongs to the interior of $\gamma_\rho$ and $\partial\gamma_\rho$ is a positively oriented closed circle. Therefore, it suffices to investigate the asymptotic properties of the right hand side (RHS) of (21). We establish the following Theorem 3.4, which provides the asymptotic behavior of eigenspace $\widehat{\boldsymbol{\Pi}}_\rho$ characterized by $\boldsymbol{u}^\mathsf{T}\widehat{\boldsymbol{\Pi}}_\rho\boldsymbol{v}$, where $\boldsymbol{u}$ and $\boldsymbol{v}$ are non-random unit vectors.

**Theorem 3.4.** *Suppose the assumptions in Lemma 2.4 hold. Let $\lambda_j^{CK}, ..., \lambda_{j+k_\rho-1}^{CK}$ be a group of isolated eigenvalues of $\boldsymbol{K}_{\mathrm{CK},\ell}$ converging to the same limit $\alpha_{\ell,1}\rho + \alpha_{\ell,0}$, where $\rho$ is defined in Theorem 3.1. For $\underline{\boldsymbol{H}}_1(\rho)$, we denote the left and right eigenvectors corresponding to 0 as $(\boldsymbol{U}_{l,\rho})_i$ and $(\boldsymbol{U}_{r,\rho})_i$ respectively, where $i$ ranges from 1 to $k_\rho$. If $\mathrm{dist}(\rho, \mathcal{S}_{1,\ell})$ is bounded away from 0, then for any non-random unit vectors $\boldsymbol{u}, \boldsymbol{v} \in \mathbb{R}^n$, we have*

$$
\boldsymbol{u}^\mathsf{T}\widehat{\boldsymbol{\Pi}}_\rho\boldsymbol{v} = -h_\ell^1(\rho)\boldsymbol{u}^\mathsf{T}\boldsymbol{Q}_1(\rho)\boldsymbol{J}\boldsymbol{F}(\rho)\boldsymbol{\Gamma}^{-1}(\rho)\boldsymbol{J}^\mathsf{T}\boldsymbol{Q}_1(\rho)\boldsymbol{v}
$$
$$
+ o_{a.s.}(1),
$$

*where*

$$
\boldsymbol{F}(\rho) = \sum_{i=1}^{k_\rho} \frac{(\boldsymbol{U}_{r,\rho})_i(\boldsymbol{U}_{l,\rho})_i^\mathsf{T}}{(\boldsymbol{U}_{l,\rho})_i^\mathsf{T}[\partial_z\underline{\boldsymbol{H}}_1(z)]_{z=\rho}(\boldsymbol{U}_{r,\rho})_i}.
$$

By noticing that $\frac{1}{p}\boldsymbol{J}^\mathsf{T}\boldsymbol{Q}_1(z)\boldsymbol{J} = \boldsymbol{\Gamma}(z)$, one may immediately derive the following corollary.

**Corollary 3.5.** *Under the same conditions as those in Theorem 3.4, we have*

$$
\frac{1}{p}\boldsymbol{J}^\mathsf{T}\widehat{\boldsymbol{\Pi}}_\rho\boldsymbol{J} = -h_\ell^1(\rho)\boldsymbol{\Gamma}(\rho)\boldsymbol{F}(\rho) + o_{a.s.}(1).
$$
(22)

This result identifies the conditions under which the eigenspace corresponding to the isolated eigenvalues of the CK matrix are informative. Discussions on non-informative eigenspace can be found in Remark 3.6 below.

*Remark* 3.6. For $\rho_+$ satisfying $\det\underline{\boldsymbol{H}}_1(\rho_+) = 0$ and $\rho_+ \to \rho \in \mathcal{H}_{1,\ell}^p$, it follows that for any non-random unit vectors $\boldsymbol{u}, \boldsymbol{v} \in \mathbb{R}^n$,
$$
\boldsymbol{u}^\mathsf{T}\widehat{\boldsymbol{\Pi}}_{\rho_+}\boldsymbol{v} = o_{a.s.}(1).
$$
(23)

When $\lambda_{j+i}$ in Equation (20) has multiplicity one, $\|\boldsymbol{\omega}_i\|$ can be regarded as a measure of the alignment between

$\hat{\boldsymbol{u}}_i$ and $\mathrm{Span}(\boldsymbol{J})$. Furthermore, if $h_\ell^1(\rho_+) \to 0$, then $\|\boldsymbol{\omega}_i\|$ approaches 0 by letting $\boldsymbol{u}, \boldsymbol{v} \in \{\boldsymbol{j}_1/\sqrt{p}, ..., \boldsymbol{j}_K/\sqrt{p}\}$, indicating that it can always be considered as a non-informative eigenvector.

The theoretical results presented above remain valid for the NTK, as summarized in Theorem 3.7 and Corollary 3.8.

**Theorem 3.7.** *Suppose the assumptions in Lemma 2.5 hold. Let $\lambda_j^{NTK}, ..., \lambda_{j+k_\rho-1}^{NTK}$ be a group of isolated eigenvalues of $\boldsymbol{K}_{\mathrm{NTK},\ell}$ that converge to the same limit $\beta_{\ell,1}\rho + \beta_{\ell,0}$, and let $\widetilde{\boldsymbol{\Pi}}_\rho$ denote the projection onto the eigenspace spanned by the eigenvectors associated with these eigenvalues. Here, $\rho$ is defined in Theorem 3.2. Suppose Assumptions 2.1-2.3 hold. For $\underline{\boldsymbol{H}}_2(\rho)$, we denote the left and right eigenvectors corresponding to 0 as $(\boldsymbol{V}_{l,z})_i$ and $(\boldsymbol{V}_{r,z})_i$, respectively, where $i = 1, ..., k_\rho$. If $\mathrm{dist}(\rho, \mathcal{S}_{2,\ell})$ is bounded away from 0, then we have*

$$
\boldsymbol{u}^\mathsf{T}\widetilde{\boldsymbol{\Pi}}_\rho\boldsymbol{v} = -h_\ell^2(\rho)\boldsymbol{u}^\mathsf{T}\boldsymbol{Q}_1(\rho)\boldsymbol{J}\widetilde{\boldsymbol{F}}(\rho)\boldsymbol{\Gamma}^{-1}(\rho)\boldsymbol{J}^\mathsf{T}\boldsymbol{Q}_1(\rho)\boldsymbol{v}
$$
$$
+ o_{a.s.}(1),
$$

*where*

$$
\widetilde{\boldsymbol{F}}(z) = \sum_{i=1}^{k_\rho} \frac{(\boldsymbol{V}_{r,\rho})_i(\boldsymbol{V}_{l,\rho})_i^\mathsf{T}}{(\boldsymbol{V}_{l,\rho})_i^\mathsf{T}[\partial_z\underline{\boldsymbol{H}}_2(z)]_{z=\rho}(\boldsymbol{V}_{r,\rho})_i}.
$$

**Corollary 3.8.** *Under the same conditions as those in Theorem 3.7, we have*

$$
\frac{1}{p}\boldsymbol{J}^\mathsf{T}\widetilde{\boldsymbol{\Pi}}_\rho\boldsymbol{J} = -\frac{1}{p}h_\ell^2(\rho)\boldsymbol{\Gamma}(\rho)\widetilde{\boldsymbol{F}} + o_{a.s.}(1).
$$

*Remark* 3.9. For the NTK, analogous to Remark 3.6, if $\rho_+$ satisfies $\det\underline{\boldsymbol{H}}_2(\rho_+) = 0$ and $\rho_+ \to \rho \in \mathcal{H}_{2,\ell}^p$, then its associated eigenspace is non-informative.

*Remark* 3.10. We note that Wang et al. (2024) investigated how spiked eigenstructures in the input data propagate through the hidden layers of a neural network by analyzing the spectrum of the CK, under the assumptions that the input data contains spiked eigenvalues and that the same activation function is applied at every layer. In contrast, our work examines both the CK and NTK. Our analysis demonstrates that isolated eigenvalues in the kernel matrices may arise from the underlying group structure of the input features, with the corresponding eigenvectors carrying useful information, even in the absence of a spiked structure in the original data as required in Wang et al. (2024). Additionally, our framework allows for different activation functions across layers and investigates their effects on the spectral properties of the kernel matrices.

### 3.3. Additional simulations and real data analysis

In this subsection, we present additional simulations on both synthetic GMM and real data to further support our theoretical findings.

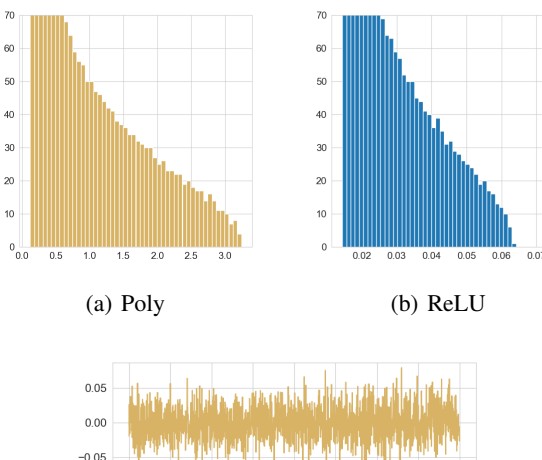

(a) Poly        (b) ReLU

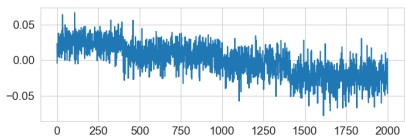

(c) Eigenvector associated with the largest eigenvalue of $\widetilde{\boldsymbol{K}}_{\mathrm{CK},3}$ when all activation functions of are Poly.

(d) Eigenvector associated with the isolated eigenvalue of $\widetilde{\boldsymbol{K}}_{\mathrm{CK},3}$ when all activation functions are ReLU.

*Figure 4.* Eigenvalue histograms ((a)-(b)) and eigenvectors ((c)-(d)) corresponding to the isolated (largest) eigenvalues of $\widetilde{\boldsymbol{K}}_{\mathrm{CK},3}$ obtained with Poly and ReLU activation functions. Parameters settings: $n = 2000$, $p = 3600$, $c_1 = c_3 = 0.2$, $c_2 = c_4 = 0.3$ and $\boldsymbol{C}_a = (1 + 8(a-1)/\sqrt{p})\boldsymbol{I}_p$ for $a = 1, 2, 3, 4$. The width $d_1, d_2, d_3$ are identical to those in Figure 1.

Figures 4(a)-(b) visualize the eigenvalues of $\widetilde{\boldsymbol{K}}_{\mathrm{CK},3}$ under different activation function settings. In Figure 4(a), all activation functions are set to Poly, consistent with those used in Figure 2, whereas in Figure 4(b), all activation functions are set to ReLU. The ReLU activation functions have been normalized and centered to satisfy the conditions specified in Assumption 2.3. Unlike the results shown in Figure 2(a), no isolated eigenvalues are observed when all activation functions are Poly. In contrast, when all activation functions are ReLU, an isolated eigenvalue emerges. Figures 4(c)-(d) show the eigenvectors corresponding to the largest eigenvalue in Figure 4(a) and the isolated eigenvalue in Figure 4(b), respectively. The former appears to be non-informative, whereas the latter is notably informative.

These observations underscore the critical role of activation function selection in determining the spectral properties of the kernel matrix.

Next, we turn to real data analysis. The input data consists

of 1600 randomly selected images from each of the digit classes 1 and 7 in the MNIST dataset, with the class-specific mean subtracted from each group. Figure 5(a) displays the spectrum of the CK, obtained using a three-layer neural network. The eigenvectors associated with the isolated eigenvalues are shown in Figure 5(b). The spectrum reveals four isolated eigenvalues, with the eigenvector corresponding to the largest eigenvalue being informative, while the other three appear non-informative. These observations are in line with our theoretical findings.

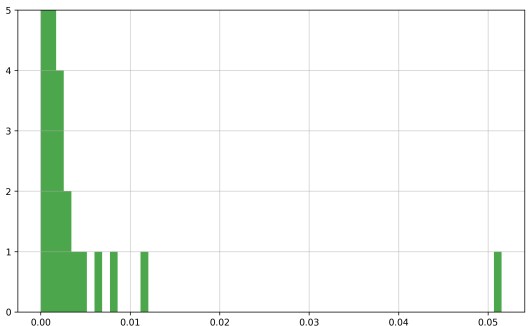

(a) Spectrum of $\widetilde{\boldsymbol{K}}_{\mathrm{CK},3}$ estimated from 500 realizations

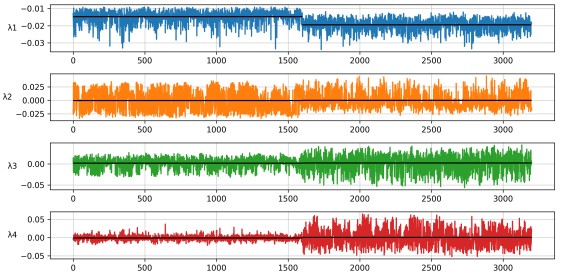

(b) Eigenvectors associated with the isolated eigenvalues.

*Figure 5.* Eigenvalue histograms (a) and eigenvectors (b) corresponding to the isolated eigenvalue of $\widetilde{\boldsymbol{K}}_{\mathrm{CK},3}$. Activations=[Sin, ReLU/10, Sin]. The weights $\boldsymbol{W}_1 \in \mathbb{R}^{d_1 \times p}$, $\boldsymbol{W}_2 \in \mathbb{R}^{d_2 \times d_1}$, $\boldsymbol{W}_3 \in \mathbb{R}^{d_3 \times d_2}$ consist of i.i.d. standard normal entries, where $d_1 = 2000, d_2 = d_3 = 1000$.

## 4. Theoretical insights for machine learning applications

It is known that the NTK theory leads to concrete convergence and generalization results (Bai & Lee, 2019) of neural networks. Let $\boldsymbol{y}$ be the true label vector of the training data and $\hat{\boldsymbol{y}}(t)$ be the prediction at time $t$. In an ultra-wide neural network with training loss $\frac{1}{2}\|\boldsymbol{y} - \hat{\boldsymbol{y}}(t)\|^2$, the time evolution of the residual $\boldsymbol{y} - \hat{\boldsymbol{y}}(t)$ during early training is approximately described by the following ordinary differential equation (Jacot et al., 2018; Du et al., 2018; 2019):

$$\frac{\mathrm{d}}{\mathrm{d}t}\hat{\boldsymbol{y}}(t) = \boldsymbol{K}_{\mathrm{NTK}}(\boldsymbol{y} - \hat{\boldsymbol{y}}(t)). \qquad (24)$$

Based on our theoretical results from Theorem 3.7–Remark 3.9, the first-order limits of entries in the isolated eigenvectors of NTK may or may not contain group features (i.e. informative or non-informative). When the eigenspace associated with the largest isolated eigenvalue contains group features, DNNs tend to prioritize learning from this subspace. Conversely, when the eigenspace lacks group features, DNNs instead prioritize learning irrelevant information, diverting attention away from effective group features. Moreover, Theorem 3.4–Remark 3.6 indicate similar phenomenon of the eigenspace during the DNN's initialization step.

## 5. Conclusion

This paper investigates the spectral properties of the conjugate kernel and neural tangent kernel, revealing the evolution of inherent group features through hidden layers. From Theorem 3.1 and Theorem 3.2, it can be seen that the occurrence of the isolated eigenvalues depends on two key factors: (1) differences in the covariance matrices between different classes and (2) the choice of activation functions. The former shapes the vector $t$ and the matrix $T$, while the latter determines the coefficients $\alpha$'s ($\beta$'s). When isolated eigenvalues are present, these factors also determine the asymptotic behaviors of the corresponding eigenvectors. This conclusion is further supported by comparing the simulation results presented in Figures 1-4.

Finally, we discuss several potential directions for future work. Since our current results rely on the assumption of GMM-distributed inputs, it is important to explore whether similar phenomena occur for more general distributions, particularly those with heavy tails. Additionally, extending our analysis to other neural network architectures represents another promising topic for future research.

## Acknowledgements

Xiangchao Li and Xiao Han are co-first authors. Qing Yang is the corresponding author. This work was supported by NSF of China (No.12371278), the CAS Talent Introduction Program (Category B), National Key R&D Program of China-2022YFA1008000, NSF of China (No.12101585) and the Young Elite Scientist Sponsorship Program by Cast (No.YESS20220125).

## Impact Statement

This paper presents work whose goal is to advance the field of Machine Learning. There are many potential societal consequences of our work, none which we feel must be specifically highlighted here.

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

## A. Appendix

In the later proof, the following lemmas are needed.

**Lemma A.1.** *(Couillet & Benaych-Georges, 2016) Suppose Assumption 2.1 holds and let $z \in \mathbb{C}$ be a point that is at least a distance $c > 0$ away from $\mathcal{S}$. We have*

$$\mathbb{E}\boldsymbol{\psi}^\mathsf{T}\boldsymbol{G}(z)\boldsymbol{\psi} = \frac{1}{p}\sum_{a=1}^{K}c_a m_a(z)\mathrm{tr}\boldsymbol{C}_a^2 + O\left(\frac{1}{p}\right), \quad \mathbb{E}|\boldsymbol{\psi}^\mathsf{T}\boldsymbol{G}(z)\boldsymbol{\psi} - \mathbb{E}\boldsymbol{\psi}^\mathsf{T}\boldsymbol{G}(z)\boldsymbol{\psi}|^m = O(p^{-\frac{m}{2}}).$$

**Lemma A.2.** *Under Assumption 2.1, for any deterministic $\boldsymbol{u} = \{u_i\}_{i=1}^{n}$ with finite Euclidean norm, we have*

$$\boldsymbol{u}^\mathsf{T}\boldsymbol{G}(z)\boldsymbol{\psi} \to 0 \quad a.s. \quad \text{for } z \text{ such that } \mathrm{dist}(\mathcal{S} \cup \{0\}, z) > c,$$

*where $c$ is a positive constant.*

We postpone the proof of Lemma A.2 to the end of this section.

### A.1. Proof of Theorem 3.1 and Theorem 3.2

Noting that the asymptotic spectral equivalents for the CK and NTK are identical up to a change of the associated coefficients $\alpha$'s to $\beta$'s, we only provide the proof for Theorem 3.1 here and the proof for Theorem 3.2 is exactly the same.

We devote the majority of this section to finding the isolated eigenvalues of $\boldsymbol{X}^\mathsf{T}\boldsymbol{X} + \alpha_{\ell,1}^{-1}\boldsymbol{V}\boldsymbol{A}_\ell\boldsymbol{V}^\mathsf{T}$, which, up to multiplication by $\alpha_{\ell,1}$ and an addition of $\alpha_{\ell,0}$, constitute the eigenvalues of $\widetilde{\boldsymbol{K}}_{\mathrm{CK},\ell}$. In order to find the isolated eigenvalues of $\boldsymbol{X}^\mathsf{T}\boldsymbol{X} + \alpha_{\ell,1}^{-1}\boldsymbol{V}\boldsymbol{A}_\ell\boldsymbol{V}^\mathsf{T}$, one only need to solve the determinant equation

$$\det(\boldsymbol{X}^\mathsf{T}\boldsymbol{X} + \alpha_{\ell,1}^{-1}\boldsymbol{V}\boldsymbol{A}_\ell\boldsymbol{V}^\mathsf{T} - z\boldsymbol{I}_n) = 0 \tag{25}$$

for $z$ away from $\mathcal{S}_{1,\ell}$. An application of Lemma A.8 yields that (25) shares the same solutions as

$$\det(\alpha_{\ell,1}^{-1}\boldsymbol{A}_\ell\boldsymbol{V}^\mathsf{T}\boldsymbol{G}(z)\boldsymbol{V} + \boldsymbol{I}_{K+1}) = 0. \tag{26}$$

According to Lemma 2.6 and the fact that

$$\frac{1}{p}\boldsymbol{J}^\mathsf{T}\boldsymbol{Q}_1(z)\boldsymbol{J} = \frac{c_0}{p}\mathrm{diag}\{n_a m_a(z)\}_{a=1}^{K} = \mathrm{diag}\{c_a m_a(z)\}_{a=1}^{K} = \boldsymbol{\Gamma}(z),$$

we have

$$\frac{1}{p}\boldsymbol{J}^\mathsf{T}\boldsymbol{G}(z)\boldsymbol{J} = \frac{1}{p}\boldsymbol{J}^\mathsf{T}\boldsymbol{Q}_1(z)\boldsymbol{J} + o_{a.s.}(1) = \boldsymbol{\Gamma}(z) + o_{a.s.}(1).$$

Lemma A.1 gives

$$\boldsymbol{\psi}^\mathsf{T}\boldsymbol{G}(z)\boldsymbol{\psi} = \frac{1}{p}\sum_{a=1}^{K}c_a m_a(z)\mathrm{tr}\boldsymbol{C}_a^2 + o_{a.s.}(1).$$

From Lemma A.2, it can be seen that the cross term $\frac{1}{\sqrt{p}}\boldsymbol{J}^\mathsf{T}\boldsymbol{G}(z)\boldsymbol{\psi}$ vanishes as $n \to \infty$. Therefore, we can write

$$\boldsymbol{V}^\mathsf{T}\boldsymbol{G}(z)\boldsymbol{V} = \begin{bmatrix} \frac{1}{p}\boldsymbol{J}^\mathsf{T}\boldsymbol{G}(z)\boldsymbol{J} & \frac{1}{\sqrt{p}}\boldsymbol{J}^\mathsf{T}\boldsymbol{G}(z)\boldsymbol{\psi} \\ \frac{1}{\sqrt{p}}\boldsymbol{\psi}^\mathsf{T}\boldsymbol{G}(z)\boldsymbol{J} & \boldsymbol{\psi}^\mathsf{T}\boldsymbol{G}(z)\boldsymbol{\psi} \end{bmatrix} = \begin{bmatrix} \boldsymbol{\Gamma}(z) & 0 \\ 0 & \frac{1}{p}\sum_{a=1}^{K}c_a m_a(z)\mathrm{tr}\boldsymbol{C}_a^2 \end{bmatrix} + o_{a.s.}(1)$$

as well as

$$\boldsymbol{I}_{K+1} + \alpha_{\ell,1}^{-1}\boldsymbol{A}_\ell\boldsymbol{V}^\mathsf{T}\boldsymbol{G}(z)\boldsymbol{V} = \boldsymbol{H}_1(z) + o_{a.s.}(1), \tag{27}$$

where

$$\boldsymbol{H}_1(z) = \begin{bmatrix} \boldsymbol{H}_{11} & \boldsymbol{H}_{12} \\ \boldsymbol{H}_{21} & \boldsymbol{H}_{22} \end{bmatrix} := \begin{bmatrix} \boldsymbol{I}_K + \alpha_{\ell,1}^{-1}[\alpha_{\ell,2}\boldsymbol{t}\boldsymbol{t}^\mathsf{T} + \alpha_{\ell,3}\boldsymbol{T}]\boldsymbol{\Gamma}(z) & [h_\ell^1(z) - 1]\boldsymbol{t} \\ \alpha_{\ell,1}^{-1}\alpha_{\ell,2}\boldsymbol{t}^\mathsf{T}\boldsymbol{\Gamma}(z) & h_\ell^1(z) \end{bmatrix}. \tag{28}$$

Therefore, it suffices to find the solution of

$$\det \boldsymbol{H}_1(z) = 0 \tag{29}$$

on the real line by (27) and Lemma A.4.

We first consider the case where $h_\ell^1(z)$ is bounded away from 0. Using the Schur complement formula, we have

$$\begin{aligned}
\det \boldsymbol{H}_1 &= \boldsymbol{H}_{22} \det \left( \boldsymbol{H}_{11} - \boldsymbol{H}_{22}^{-1} \boldsymbol{H}_{12} \boldsymbol{H}_{21} \right) \\
&= \boldsymbol{H}_{22}^{1-K} \det(\boldsymbol{H}_{11} \boldsymbol{H}_{22} - \boldsymbol{H}_{12} \boldsymbol{H}_{21}) = \boldsymbol{H}_{22}^{1-K} \det \underline{\boldsymbol{H}}_1.
\end{aligned} \tag{30}$$

Therefore, if $\rho$ satisfies that $\mathrm{dist}(\rho, \mathcal{S}_{1,\ell}) > C$ for some constant $C$ and makes $\underline{\boldsymbol{H}}_1(\rho)$ a singular matrix, with the multiplicity of the eigenvalue 0 denoted by $k_\rho$, then according to (5), there exists at least one eigenvalue of $\boldsymbol{X}^\top \boldsymbol{X} + \alpha_{\ell,1}^{-1} \boldsymbol{V} \boldsymbol{A}_\ell \boldsymbol{V}^\top$ that converges to $\rho$ almost surely.

If $h_\ell^1(z) \to 0$, studying (30) is not suitable because $\boldsymbol{H}_{22}^{1-K} = h_\ell^1(z)^{1-K}$ tends to infinity. We instead consider the matrix $\boldsymbol{H}_1(z)$ directly. If (29) has a solution $\rho_+$ such that $\rho_+ \to \rho \in \mathcal{H}_{1,\ell}^p$, then $\rho$ is a deterministic limit of certain eigenvalues of $\widetilde{\boldsymbol{K}}_{\mathrm{CK},\ell}$.

Next, we investigate the number of eigenvalues of $\boldsymbol{K}_{\mathrm{CK},\ell}$ that converge to the limits identified above. By Lemma 2.6, we see that the eigenvalues of $\boldsymbol{X}^\top \boldsymbol{X}$ asymptotically do not escape $\mathcal{S} \cup \{0\}$. Therefore, one may find a compact interval $\mathcal{I} = [a_1, b_1] \subset (a, b)$ such that, for large enough $n$, $\boldsymbol{X}^\top \boldsymbol{X}$ has no eigenvalue in $\mathcal{I}$ with probability one and

$$\{\rho \mid \det \boldsymbol{H}_1(\rho) = 0\} \cap \{a_1, b_1\} = \emptyset. \tag{31}$$

We use $D_\mathcal{I}^\circ$ to denote the open disc centered at $(a_1 + b_1)/2$ and of diameter $(b_1 - a_1)$. Let

$$L_1 := \#\{\text{zeros of } \det(\boldsymbol{X}^\top \boldsymbol{X} + \alpha_{\ell,1}^{-1} \boldsymbol{V} \boldsymbol{A}_\ell \boldsymbol{V}^\top - z \boldsymbol{I}_n) \text{ in } D_\mathcal{I}^\circ\}, \quad L_2 := \#\{\rho \mid \rho \in D_\mathcal{I}^\circ \text{ and } \det \boldsymbol{H}_1(\rho) = 0\}.$$

Note that the functions $\det(\boldsymbol{X}^\top \boldsymbol{X} + \alpha_{\ell,1}^{-1} \boldsymbol{V} \boldsymbol{A}_\ell \boldsymbol{V}^\top - z \boldsymbol{I}_n)$, $\det \boldsymbol{G}(z)$ and $\det \boldsymbol{H}_1(z)$ are analytic on $D_\mathcal{I}^\circ$ and non-vanishing on $\mathbb{C} \backslash \mathbb{R}$. By Lemma A.5 and the argument principle, we obtain

$$\begin{aligned}
L_1 &= \frac{1}{2\pi \mathrm{i}} \oint_{\partial D_\mathcal{I}^\circ} \frac{\partial_z \det(\boldsymbol{X}^\top \boldsymbol{X} + \alpha_{\ell,1}^{-1} \boldsymbol{V} \boldsymbol{A}_\ell \boldsymbol{V}^\top - z \boldsymbol{I}_n)}{\det(\boldsymbol{X}^\top \boldsymbol{X} + \alpha_{\ell,1}^{-1} \boldsymbol{V} \boldsymbol{A}_\ell \boldsymbol{V}^\top - z \boldsymbol{I}_n)} \\
&= \frac{1}{2\pi \mathrm{i}} \oint_{\partial D_\mathcal{I}^\circ} \frac{\partial_z \det(\boldsymbol{I}_n + \alpha_{\ell,1}^{-1} \boldsymbol{V} \boldsymbol{A}_\ell \boldsymbol{V}^\top \boldsymbol{G}(z))}{\det(\boldsymbol{I}_n + \alpha_{\ell,1}^{-1} \boldsymbol{V} \boldsymbol{A}_\ell \boldsymbol{V}^\top \boldsymbol{G}(z))} + \frac{1}{2\pi \mathrm{i}} \oint_{\partial D_\mathcal{I}^\circ} \frac{\partial_z \det \boldsymbol{G}(z)}{\det \boldsymbol{G}(z)} \\
&= \frac{1}{2\pi \mathrm{i}} \oint_{\partial D_\mathcal{I}^\circ} \frac{\partial_z \det(\boldsymbol{I}_{K+1} + \alpha_{\ell,1}^{-1} \boldsymbol{A}_\ell \boldsymbol{V}^\top \boldsymbol{G}(z) \boldsymbol{V})}{\det(\boldsymbol{I}_{K+1} + \alpha_{\ell,1}^{-1} \boldsymbol{A}_\ell \boldsymbol{V}^\top \boldsymbol{G}(z) \boldsymbol{V})} \to \frac{1}{2\pi \mathrm{i}} \oint_{\partial D_\mathcal{I}^\circ} \frac{\partial_z \det \boldsymbol{H}_1(z)}{\det \boldsymbol{H}_1(z)} = L_2
\end{aligned}$$

almost surely, where $\partial D_\mathcal{I}^\circ$ is seen as a positively oriented contour. Since both $L_1$ and $L_2$ are integers, the multiplicity of an isolated eigenvalue of $\boldsymbol{X}^\top \boldsymbol{X} + \alpha_{\ell,1}^{-1} \boldsymbol{V} \boldsymbol{A}_\ell \boldsymbol{V}^\top$, which has a deterministic limit $\rho$, is the same as that of 0 as an eigenvalue of $\boldsymbol{H}_1(\rho)$. It is straightforward to see that if $\rho_+$ defined in Theorem 3.1 satisfies $h_\ell^1(\rho_+) \to 0$, then the multiplicity of $\rho_+$ denoted by $k_\rho^+$ is equal to the multiplicity of 0 as an eigenvalue of $\boldsymbol{H}_1(\rho_+)$. In the case that $h_\ell^1(\rho)$ is away from 0, we have that $\underline{\boldsymbol{H}}_1(\rho)$ has a 0 eigenvalue of multiplicity $k_\rho$ by (30). This concludes the proof of Theorem 3.1.

## A.2. Proof of Theorem 3.4 and Theorem 3.7

For any closed interval $\mathcal{I} \subset \mathbb{R}$ defined above (31), we have

$$-\frac{1}{2\pi \mathrm{i}} \oint_{\partial D_\mathcal{I}^\circ} \boldsymbol{u}^\top \boldsymbol{G}(z) \boldsymbol{v} \mathrm{d}z = 0$$

almost surely. Lemma A.2 implies that, for any deterministic $\boldsymbol{u}$ and $\boldsymbol{v}$ with finite Euclidean norm,

$$\boldsymbol{u}^\top \boldsymbol{G}(z) \boldsymbol{V} = \boldsymbol{u}^\top \boldsymbol{Q}_1(z) \boldsymbol{V} + o_{a.s.}(1), \quad \boldsymbol{v}^\top \boldsymbol{G}(z) \boldsymbol{V} = \boldsymbol{v}^\top \boldsymbol{Q}_1(z) \boldsymbol{V} + o_{a.s.}(1).$$

Then by using Woodbury's identity, we obtain

$$
\begin{aligned}
\boldsymbol{u}^\mathsf{T}\widehat{\boldsymbol{\Pi}}_\mathcal{I}\boldsymbol{v} &= -\frac{1}{2\pi\mathrm{i}}\oint_{\partial D_\mathcal{I}^o}\boldsymbol{u}^\mathsf{T}(\boldsymbol{X}^\mathsf{T}\boldsymbol{X}+\alpha_{\ell,1}^{-1}\boldsymbol{V}\boldsymbol{A}_\ell\boldsymbol{V}^\mathsf{T}-z\boldsymbol{I}_n)^{-1}\boldsymbol{v}\mathrm{d}z \\
&= -\frac{1}{2\pi\mathrm{i}}\oint_{\partial D_\mathcal{I}^o}\boldsymbol{u}^\mathsf{T}\boldsymbol{G}(z)\boldsymbol{v}\mathrm{d}z \\
&\quad + \frac{1}{2\pi\mathrm{i}}\oint_{\partial D_\mathcal{I}^o}\alpha_{\ell,1}^{-1}\boldsymbol{u}^\mathsf{T}\boldsymbol{G}(z)\boldsymbol{V}(\boldsymbol{I}_{k+1}+\alpha_{\ell,1}^{-1}\boldsymbol{A}_\ell\boldsymbol{V}^\mathsf{T}\boldsymbol{G}(z)\boldsymbol{V})^{-1}\boldsymbol{A}_\ell\boldsymbol{V}^\mathsf{T}\boldsymbol{G}(z)\boldsymbol{v}\mathrm{d}z \\
&= \frac{1}{2\pi\mathrm{i}}\oint_{\partial D_\mathcal{I}^o}\alpha_{\ell,1}^{-1}\boldsymbol{u}^\mathsf{T}\boldsymbol{Q}_1(z)\boldsymbol{V}\boldsymbol{H}_1^{-1}(z)\boldsymbol{A}_\ell\boldsymbol{V}^\mathsf{T}\boldsymbol{Q}_1(z)\boldsymbol{v}\mathrm{d}z + o_{a.s.}(1),
\end{aligned}
\tag{32}
$$

where $\widehat{\boldsymbol{\Pi}}_\mathcal{I}$ is the projection onto the eigenspace spanned by the eigenvectors corresponding to the eigenvalues of $\boldsymbol{K}_{\mathrm{CK},\ell}$ that lie inside $\mathcal{I}$. Thus what remains to find the limit of (32) is to obtain the deterministic equivalent of $\boldsymbol{H}_1^{-1}(z)\boldsymbol{A}_\ell$. Since the zeros of $h_\ell(z)$ are away from $\bar{D}_\mathcal{I}^o$, we obtain

$$
\boldsymbol{H}_1^{-1}(z) = \begin{bmatrix} \boldsymbol{H}_{22}\underline{\boldsymbol{H}}_1^{-1}(z) & -\underline{\boldsymbol{H}}_1^{-1}(z)\boldsymbol{H}_{12} \\ -\boldsymbol{H}_{12}\underline{\boldsymbol{H}}_1^{-1} & \boldsymbol{H}_{22}^{-1}+\boldsymbol{H}_{22}^{-1}\boldsymbol{H}_{21}\underline{\boldsymbol{H}}_1^{-1}(z)\boldsymbol{H}_{12} \end{bmatrix}.
$$

All notations presented here are defined in (28). Therefore, we can write

$$
\boldsymbol{H}_1^{-1}(z)\boldsymbol{A}_\ell = \begin{bmatrix} [\boldsymbol{H}_1^{-1}(z)\boldsymbol{A}_\ell]_{11} & [\boldsymbol{H}_1^{-1}(z)\boldsymbol{A}_\ell]_{12} \\ [\boldsymbol{H}_1^{-1}(z)\boldsymbol{A}_\ell]_{21} & [\boldsymbol{H}_1^{-1}(z)\boldsymbol{A}_\ell]_{22} \end{bmatrix},
$$

where

$$
[\boldsymbol{H}_1^{-1}(z)\boldsymbol{A}_\ell]_{11} = \alpha_{\ell,3}h_\ell^1(z)\underline{\boldsymbol{H}}_1^{-1}(z)\boldsymbol{T}+\alpha_{\ell,2}\underline{\boldsymbol{H}}_1^{-1}(z)\boldsymbol{t}\boldsymbol{t}^\mathsf{T},
$$

$$
[\boldsymbol{H}_1^{-1}(z)\boldsymbol{A}_\ell]_{12} = \alpha_{\ell,2}\underline{\boldsymbol{H}}_1^{-1}(z)\boldsymbol{t},
$$

$$
[\boldsymbol{H}_1^{-1}(z)\boldsymbol{A}_\ell]_{21} = -\frac{\alpha_{\ell,2}\alpha_{\ell,3}}{\alpha_{\ell,1}}\boldsymbol{t}^\mathsf{T}\boldsymbol{\Gamma}(z)\underline{\boldsymbol{H}}_1^{-1}(z)\boldsymbol{T}+\frac{\alpha_{\ell,2}\boldsymbol{t}^\mathsf{T}}{h_\ell^1(z)}+\frac{1}{h_\ell^1(z)}\frac{\alpha_{\ell,2}^2}{\alpha_{\ell,1}}\boldsymbol{t}^\mathsf{T}\boldsymbol{\Gamma}(z)\underline{\boldsymbol{H}}_1^{-1}(z)\boldsymbol{t}\boldsymbol{t}^\mathsf{T},
$$

$$
[\boldsymbol{H}_1^{-1}(z)\boldsymbol{A}_\ell]_{22} = \frac{\alpha_{\ell,2}}{h_\ell^1(z)}-\frac{1}{h_\ell^1(z)}\frac{\alpha_{\ell,2}^2}{\alpha_{\ell,1}}\boldsymbol{t}^\mathsf{T}\boldsymbol{\Gamma}(z)\underline{\boldsymbol{H}}_1^{-1}(z)\boldsymbol{t}.
$$

Observing that $\boldsymbol{u}^\mathsf{T}\boldsymbol{G}(z)\boldsymbol{V} = \frac{1}{\sqrt{p}}[\boldsymbol{u}^\mathsf{T}\boldsymbol{Q}_1(z)\boldsymbol{J}\quad 0]+o_{a.s.}(1)$ $(\boldsymbol{v}^\mathsf{T}\boldsymbol{G}(z)\boldsymbol{V} = \frac{1}{\sqrt{p}}[\boldsymbol{v}^\mathsf{T}\boldsymbol{Q}_1(z)\boldsymbol{J}\quad 0]+o_{a.s.}(1)$ ), it becomes evident that only $[\boldsymbol{H}_1^{-1}(z)\boldsymbol{A}_\ell]_{11}$ is needed. Then we deduce that

$$
\begin{aligned}
&\alpha_{\ell,1}^{-1}\boldsymbol{u}^\mathsf{T}\boldsymbol{Q}(z)\boldsymbol{V}\boldsymbol{H}_1^{-1}(z)\boldsymbol{A}_\ell\boldsymbol{V}^\mathsf{T}\boldsymbol{Q}(z)\boldsymbol{v} \\
=&\frac{1}{p}\alpha_{\ell,1}^{-1}\boldsymbol{u}^\mathsf{T}\boldsymbol{Q}_1(z)\boldsymbol{J}[\alpha_{\ell,3}h_\ell^1(z)\underline{\boldsymbol{H}}_1^{-1}(z)\boldsymbol{T}+\alpha_{\ell,2}\underline{\boldsymbol{H}}_1^{-1}(z)\boldsymbol{t}\boldsymbol{t}^\mathsf{T}]\boldsymbol{J}\boldsymbol{Q}_1(z)\boldsymbol{v} + o_{a.s.}(1).
\end{aligned}
$$

We denote the spectral decomposition of $\underline{\boldsymbol{H}}_1(z)$ as

$$
\underline{\boldsymbol{H}}_1(z) = \boldsymbol{U}_{r,z}\Lambda_z\boldsymbol{U}_{l,z}^\mathsf{T}.
$$

For $\rho\in\mathcal{I}$ such that $\underline{\boldsymbol{H}}_1(\rho)$ is singular and $k_\rho$ denotes the multiplicity of 0 as an eigenvalue of $\underline{\boldsymbol{H}}_1(\rho)$, we denote the left and right eigenvectors corresponding to 0 as $(\boldsymbol{U}_{l,\rho})_i$ and $(\boldsymbol{U}_{r,\rho})_i$ respectively, where $i$ ranges from 1 to $k_\rho$. For a matrix $\boldsymbol{M}(z) = \{M_{ij}(z)\}_{i,j=1}^K$, denote $\mathrm{Res}(\boldsymbol{M}) = \{\mathrm{Res}(M_{ij}(z))\}_{i,j=1}^K$. Then we have

$$
\begin{aligned}
\lim_{z\to\rho}(z-\rho)\underline{\boldsymbol{H}}_1^{-1}(z) &= \lim_{z\to\rho}(z-\rho)\boldsymbol{U}_{r,z}\Lambda_z^{-1}\boldsymbol{U}_{l,z}^\mathsf{T} \\
&= \lim_{z\to\rho}(z-\rho)\sum_{i=1}^{k_\rho}\frac{(\boldsymbol{U}_{r,z})_i(\boldsymbol{U}_{l,z})_i^\mathsf{T}}{(\boldsymbol{U}_{l,z})_i^\mathsf{T}\underline{\boldsymbol{H}}_1(z)(\boldsymbol{U}_{r,z})_i} \\
&= \lim_{z\to\rho}\sum_{i=1}^{k_\rho}\frac{(\boldsymbol{U}_{r,z})_i(\boldsymbol{U}_{l,z})_i^\mathsf{T}}{\partial_z(\boldsymbol{U}_{l,z})_i^\mathsf{T}\underline{\boldsymbol{H}}_1(z)(\boldsymbol{U}_{r,z})_i},
\end{aligned}
$$

where the last equation is obtained by applying the L'Hôpital's rule. Since

$$(\boldsymbol{U}_{l,\rho})_i^\mathsf{T} \underline{\boldsymbol{H}}_1(\rho) = 0 \quad \text{and} \quad \underline{\boldsymbol{H}}_1(\rho)(\boldsymbol{U}_{r,\rho})_i = 0,$$

we obtain

$$\begin{aligned}
&[\partial_z (\boldsymbol{U}_{l,z})_i^\mathsf{T} \underline{\boldsymbol{H}}_1(z)(\boldsymbol{U}_{r,z})_i]_{z=\rho} \\
&= [\partial_z (\boldsymbol{U}_{l,z})_i^\mathsf{T}]_{z=\rho} \underline{\boldsymbol{H}}_1(z)(\boldsymbol{U}_{r,z})_i + (\boldsymbol{U}_{l,z})_i^\mathsf{T} [\partial_z \underline{\boldsymbol{H}}_1(z)]_{z=\rho}(\boldsymbol{U}_{r,z})_i + (\boldsymbol{U}_{l,z})_i^\mathsf{T} \underline{\boldsymbol{H}}_1(z)[\partial_z ((\boldsymbol{U}_{r,z})_i)]_{z=\rho} \\
&= (\boldsymbol{U}_{l,z})_i^\mathsf{T} [\partial_z \underline{\boldsymbol{H}}_1(z)]_{z=\rho}(\boldsymbol{U}_{r,z})_i.
\end{aligned}$$

It is easy to see that

$$\begin{aligned}
&\frac{1}{p} \boldsymbol{u}^\mathsf{T} \boldsymbol{Q}_1(z) \boldsymbol{J} \left\{ \underline{\boldsymbol{H}}_1^{-1}(z) \left[ \frac{\alpha_{\ell,3}}{\alpha_{\ell,1}} h_\ell^1(z) \boldsymbol{T} + \frac{\alpha_{\ell,2}}{\alpha_{\ell,1}} \boldsymbol{t} \boldsymbol{t}^\mathsf{T} \right] \right\} \boldsymbol{J}^\mathsf{T} \boldsymbol{Q}_1(z) \boldsymbol{v} \\
&= \frac{1}{p} \boldsymbol{u}^\mathsf{T} \boldsymbol{Q}_1(z) \boldsymbol{J} \left\{ \underline{\boldsymbol{H}}_1^{-1}(z) \left[ \frac{\alpha_{\ell,3}}{\alpha_{\ell,1}} h_\ell^1(z) \boldsymbol{T} + \frac{\alpha_{\ell,2}}{\alpha_{\ell,1}} \boldsymbol{t} \boldsymbol{t}^\mathsf{T} \right] \boldsymbol{\Gamma}(z) \boldsymbol{\Gamma}^{-1}(z) \right\} \boldsymbol{J}^\mathsf{T} \boldsymbol{Q}_1(z) \boldsymbol{v} \\
&= \frac{1}{p} \boldsymbol{u}^\mathsf{T} \boldsymbol{Q}_1(z) \boldsymbol{J} \left\{ \underline{\boldsymbol{H}}_1^{-1}(z) [\boldsymbol{H}_1(z) - h_\ell^1(z)] \boldsymbol{\Gamma}^{-1}(z) \right\} \boldsymbol{J}^\mathsf{T} \boldsymbol{Q}_1(z) \boldsymbol{v} \\
&= \frac{1}{p} \boldsymbol{u}^\mathsf{T} \boldsymbol{Q}_1(z) \boldsymbol{J} [\boldsymbol{\Gamma}^{-1}(z) - h_\ell^1(z) \underline{\boldsymbol{H}}_1^{-1}(z) \boldsymbol{\Gamma}^{-1}(z)] \boldsymbol{J}^\mathsf{T} \boldsymbol{Q}_1(z) \boldsymbol{v}.
\end{aligned}$$

Now one can conclude by (32) that

$$\begin{aligned}
\boldsymbol{u}^\mathsf{T} \widehat{\boldsymbol{\Pi}}_\mathcal{I} \boldsymbol{v} &= -\frac{1}{p} \sum_{z \in \mathcal{I}} \mathrm{Res} \left( h_\ell^1(z) \boldsymbol{u}^\mathsf{T} \boldsymbol{Q}_1(z) \boldsymbol{J} \underline{\boldsymbol{H}}_1^{-1}(z) \boldsymbol{J}^\mathsf{T} \boldsymbol{Q}_1(z) \boldsymbol{v} \right) + o_{a.s.}(1) \\
&= -\frac{1}{p} h_\ell^1(\rho) \boldsymbol{u}^\mathsf{T} \boldsymbol{Q}_1(\rho) \boldsymbol{J} \sum_{i=1}^{k_\rho} \frac{(\boldsymbol{U}_{r,\rho})_i (\boldsymbol{U}_{l,\rho})_i^\mathsf{T}}{(\boldsymbol{U}_{l,\rho})_i^\mathsf{T} [\partial_z \underline{\boldsymbol{H}}_1(z)]_{z=\rho} (\boldsymbol{U}_{r,\rho})_i} \boldsymbol{J}^\mathsf{T} \boldsymbol{Q}_1(\rho) \boldsymbol{v} + o_{a.s.}(1).
\end{aligned}$$

Thus the proof of Theorem 3.4 is completed. Theorem 3.7 can be proved by the same lines as that of Theorem 3.4.

### A.3. Proof of Equation (23)

Now, we consider the case that $\rho_+ \to \rho \in \mathcal{H}_{1,\ell}^p$, with $\rho$ being at a distance from $\mathcal{S}_\ell$ and $\det \boldsymbol{H}_1(\rho_+) = 0$. Suppose $\|\boldsymbol{t}\|$ is bounded away from 0, we note that $\lim_{z \to \rho} \underline{\boldsymbol{H}}_1(z) = \frac{\alpha_{\ell,2}}{\alpha_{\ell,1}} \boldsymbol{t} \boldsymbol{t}^\mathsf{T} \boldsymbol{\Gamma}(z)$ and the RHS is a rank-1 matrix. Hence there are $K-1$ eigenvalues of $\underline{\boldsymbol{H}}_1(\rho_+)$ converging to 0 almost surely. It is natural to see that the derivative $\partial_z \underline{\boldsymbol{H}}_1(z)|_{z=\rho_+}$ is well defined and not close to 0. Based on the above facts and applying the L'Hôpital's rule again, we have

$$\begin{aligned}
\lim_{z \to \rho_+} (z - \rho_+) h_\ell^1(z) \underline{\boldsymbol{H}}_1^{-1}(z) &= \lim_{z \to \rho_+} (z - \rho_+) h_\ell^1(z) \boldsymbol{U}_{r,z} \boldsymbol{\Lambda}_z^{-1} \boldsymbol{U}_{l,z}^\mathsf{T} \\
&= \lim_{z \to \rho_+} h_\ell^1(z) \lim_{z \to \rho_+} \sum_{i=1}^{K-1} (z - \rho_+) \frac{(\boldsymbol{U}_{r,\rho})_i (\boldsymbol{U}_{l,\rho})_i^\mathsf{T}}{(\boldsymbol{U}_{l,\rho_+})_i^\mathsf{T} \boldsymbol{H}_1(z) (\boldsymbol{U}_{r,\rho_+})_i} \\
&= \lim_{z \to \rho_+} h_\ell^1(z) \sum_{i=1}^{K-1} \frac{(\boldsymbol{U}_{r,\rho_+})_i (\boldsymbol{U}_{l,\rho_+})_i^\mathsf{T}}{(\boldsymbol{U}_{l,\rho_+})_i^\mathsf{T} [\partial_z \underline{\boldsymbol{H}}_1(z)]_{z=\rho_+} (\boldsymbol{U}_{r,\rho_+})_i} \\
&= 0.
\end{aligned}$$

If $\|\boldsymbol{t}\| \to 0$, then $\rho_+$ becomes a removable singularity for $h_\ell^1(z) \underline{\boldsymbol{H}}_1^{-1}(z)$, thereby ensuring that the residue remains zero. The conclusion can be obtained by repeating the proof of Theorem 3.4.

### A.4. Proof of Lemma A.2

We divide the proof into three steps:

**Step 1**: Fix $\eta > 0$. For any $z \in \mathbb{C}$ with $\Im z \geq \eta$, the expectation $\mathbb{E} \boldsymbol{u}^\mathsf{T} \boldsymbol{G}(z) \boldsymbol{\psi} = o(1)$.

**Step 2**: Convergence of $\boldsymbol{u}^\mathsf{T}\boldsymbol{G}(z)\boldsymbol{\psi}$ to $\mathbb{E}\boldsymbol{u}^\mathsf{T}\boldsymbol{G}(z)\boldsymbol{\psi}$ almost surely.

**Step 3**: For any $z$ such that $\mathrm{dist}(\mathcal{S} \cup \{0\}, z) > c$ for some constant $c$, we also have $\boldsymbol{u}^\mathsf{T}\boldsymbol{G}(z)\boldsymbol{\psi} \to 0$ $a.s.$

We assume $\Im z \geq \eta > 0$ if $\Re z = E > 0$. For the case $\Im z \leq -\eta < 0$, one only needs to replace $\eta$ by $|\eta|$ in the following proof. Define

$$\hat{\boldsymbol{X}}_j = \boldsymbol{X} - \boldsymbol{x}_j\boldsymbol{e}_j^\mathsf{T}, \quad \hat{\boldsymbol{G}}_j(z) = (\hat{\boldsymbol{X}}_j^\mathsf{T}\hat{\boldsymbol{X}}_j - z\boldsymbol{I}_n)^{-1},$$

where $\boldsymbol{e}_j \in \mathbb{R}^n$ denotes the vector with the $k$th element being 1 and otherwise being 0. Note that $\boldsymbol{\psi} = \sum_{j=1}^n \psi_j\boldsymbol{e}_j$. We can write

$$\boldsymbol{u}^\mathsf{T}\boldsymbol{G}(z)\boldsymbol{\psi} = \sum_{j=1}^n \boldsymbol{u}^\mathsf{T}\boldsymbol{G}(z)\psi_j\boldsymbol{e}_j.$$

Decompose the term $\mathbb{E}\boldsymbol{u}^\mathsf{T}\boldsymbol{G}(z)\psi_j\boldsymbol{e}_j$ as

$$\boldsymbol{u}^\mathsf{T}\mathbb{E}\boldsymbol{G}(z)\psi_j\boldsymbol{e}_j = \boldsymbol{u}^\mathsf{T}\mathbb{E}[\boldsymbol{G}(z) - \hat{\boldsymbol{G}}_j(z)]\psi_j\boldsymbol{e}_j + \boldsymbol{u}^\mathsf{T}\mathbb{E}\hat{\boldsymbol{G}}_j(z)\psi_j\boldsymbol{e}_j.$$

Since $\hat{\boldsymbol{G}}_j(z)$ is independent of $\psi_j$, we have $\mathbb{E}\hat{\boldsymbol{G}}_j(z)\psi_j\boldsymbol{e}_j = \mathbb{E}\hat{\boldsymbol{G}}_j(z)\mathbb{E}\psi_j\boldsymbol{e}_j = 0$. Let

$$\boldsymbol{X}_j = [\boldsymbol{x}_1, ..., \boldsymbol{x}_{j-1}, \boldsymbol{x}_{j+1}, ..., \boldsymbol{x}_n], \quad \boldsymbol{G}_j(z) = (\boldsymbol{X}_j^\mathsf{T}\boldsymbol{X}_j - z\boldsymbol{I}_{n-1})^{-1},$$

$$\beta_j = (\boldsymbol{x}_j^\mathsf{T}\boldsymbol{x}_j - z - \boldsymbol{x}_j^\mathsf{T}\hat{\boldsymbol{X}}_j\boldsymbol{G}_j(z)\boldsymbol{X}_j^\mathsf{T}\boldsymbol{x}_j)^{-1}, \quad b_j = \left(\frac{\mathrm{tr}\boldsymbol{C}_{g(j)}}{p} - z - \frac{\mathbb{E}\mathrm{tr}\boldsymbol{C}_{g(j)}\boldsymbol{X}_j\boldsymbol{G}_j(z)\boldsymbol{X}_j^\mathsf{T}}{p}\right)^{-1},$$

where $g(j) \in \{1, ..., K\}$ denotes the class to which $\boldsymbol{x}_j$ belongs. All of these quantities are bounded in absolute value by $\eta^{-1}$. Then from Lemma A.6, we get for $m \geq 2$,

$$\mathbb{E}|\beta_j - b_j|^m \leq \frac{1}{\eta^{2m}}\mathbb{E}\left|\boldsymbol{x}_j^\mathsf{T}\boldsymbol{x}_j - \boldsymbol{x}_j^\mathsf{T}\boldsymbol{X}_j\boldsymbol{G}_j(z)\boldsymbol{X}_j^\mathsf{T}\boldsymbol{x}_j - \left(\frac{\mathrm{tr}\boldsymbol{C}_{g(j)}}{p} - \frac{\mathbb{E}\mathrm{tr}\boldsymbol{C}_{g(j)}\boldsymbol{X}_j\boldsymbol{G}_j(z)\boldsymbol{X}_j^\mathsf{T}}{p}\right)\right|^m \tag{33}$$
$$= O\left((p\eta^6)^{-\frac{m}{2}}\right),$$

and

$$\mathbb{E}\psi_j^m = \frac{1}{p^m}\mathbb{E}(\boldsymbol{x}_j^\mathsf{T}\boldsymbol{x}_j - \mathrm{tr}\boldsymbol{C}_{g(j)})^m = O\left(p^{-\frac{m}{2}}\right), \quad m \geq 2. \tag{34}$$

By the Schur complement formula, we obtain

$$\psi_j\boldsymbol{u}^\mathsf{T}[\boldsymbol{G}(z) - \hat{\boldsymbol{G}}_j(z)]\boldsymbol{e}_j = -\psi_j[(\beta_j - z^{-1})u_j - \beta_j\boldsymbol{u}_j^\mathsf{T}\boldsymbol{G}_j(z)\boldsymbol{X}_j^\mathsf{T}\boldsymbol{x}_j],$$

where $\boldsymbol{u}_j = [u_1, ..., u_{j-1}, u_{j+1}, ..., u_n]$. It follows from (33) and (34) that

$$|\mathbb{E}\psi_j\beta_j u_j| = |\mathbb{E}\psi_j(\beta_j - b_j)u_j| \leq |u_j|\mathbb{E}|\psi_j(\beta_j - b_j)|$$
$$\leq |u_j|\sqrt{\mathbb{E}|\psi_j|^2\mathbb{E}|\beta_j - b_j|^2} = |u_j|O\left(\frac{1}{\eta^3 p}\right).$$

Thus we have

$$\sum_{j=1}^n \mathbb{E}\psi_j\beta_j u_j \leq \sum_{j=1}^n |\mathbb{E}\psi_j\beta_j u_j| \leq \frac{c}{\eta^3 p}\sum_{j=1}^n |u_j| = O\left(\frac{1}{\eta^3 p^{1/2}}\right) \to 0. \tag{35}$$

We write $\boldsymbol{x}_j = \frac{1}{\sqrt{p}}\boldsymbol{C}_{g(j)}^{1/2}\boldsymbol{z}_j$, where $\boldsymbol{z}_j = \{z_i^j\}_{i=1}^p$ is a random vector with i.i.d. standard Gaussian entries. Denote $\boldsymbol{y}_j^\mathsf{T} = \boldsymbol{u}_j^\mathsf{T}\boldsymbol{G}_j(z)\boldsymbol{X}_j^\mathsf{T}\boldsymbol{C}_{g(j)}^{1/2} = \{y_i^j\}_{i=1}^p$. It is easy to see that $\|\boldsymbol{y}_j\| \leq \|\boldsymbol{u}_j\|\|\boldsymbol{G}_j(z)\boldsymbol{X}_j\|\|\boldsymbol{C}_{g(j)}^{1/2}\| = O(\eta^{-1})$. For $m \geq 1$, by Lemma A.7, it is true that

$$\mathbb{E}|\boldsymbol{u}_j^\mathsf{T}\boldsymbol{G}_j\boldsymbol{X}_j^\mathsf{T}\boldsymbol{x}_j|^{2m} = \mathbb{E}\left|\frac{1}{\sqrt{p}}\boldsymbol{y}_j^\mathsf{T}\boldsymbol{z}_j\right|^{2m} = \frac{1}{p^m}\mathbb{E}\mathbb{E}[|\boldsymbol{y}_j^\mathsf{T}\boldsymbol{z}_j|^{2m}|\boldsymbol{y}_j] = \frac{1}{p^m}\mathbb{E}\mathbb{E}\left[\left|\sum_{i=1}^p y_i^j z_i^j\right|^{2m}\bigg|\boldsymbol{y}_j\right]$$
$$\leq \mathbb{E}\left\{c\left(\sum_{i=1}^p \frac{|y_i^j|^2}{p}\right)^m + c\mathbb{E}\left[\sum_{i=1}^p \left|y_i^j\frac{z_i^j}{\sqrt{p}}\right|^{2m}\bigg|\boldsymbol{y}_j\right]\right\} \tag{36}$$
$$\leq \frac{C}{\eta^{2m}p^m} + C\mathbb{E}\frac{\sum_{i=1}^p |y_i^j|^{2m}}{p^m} = O\left(\frac{1}{\eta^{2m}p^m}\right),$$

where we use the inequality

$$\sum_{i=1}^{p} |y_i^j|^{2m} \le \left( \sum_{i=1}^{p} |y_i^j|^2 \right)^m = \|\boldsymbol{y}_j\|^{2m} = O(\eta^{-2m}).$$

Write $\mathbb{E}\psi_j \boldsymbol{u}_j^\top \beta_j \boldsymbol{G}_j(z) \boldsymbol{X}_j^\top \boldsymbol{x}_j = \mathbb{E}\psi_j \boldsymbol{u}_j^\top (\beta_j - b_j) \boldsymbol{G}_j(z) \boldsymbol{X}_j^\top \boldsymbol{x}_j$. It can be seen from (33), (34) and (36) that

$$\begin{aligned}
|\mathbb{E}\psi_j \boldsymbol{u}_j^\top \beta_j \boldsymbol{G}_j(z) \boldsymbol{X}_j^\top \boldsymbol{x}_j| &\le \mathbb{E}|\psi_j \boldsymbol{u}_j^\top (\beta_j - b_j) \boldsymbol{G}_j(z) \boldsymbol{X}_j^\top \boldsymbol{x}_j| \le \mathbb{E}|\psi_j||\beta_j - b_j||\boldsymbol{u}_j^\top \boldsymbol{G}_j(z) \boldsymbol{X}_j^\top \boldsymbol{x}_j| \\
&\le (\mathbb{E}|\psi_j|^4 \mathbb{E}|\beta_j - b_j|^4)^{\frac{1}{4}} \sqrt{\mathbb{E}|\boldsymbol{y}_j^\top \boldsymbol{x}_j|^2} \\
&\le \frac{c}{\eta^4 p} \sqrt{\mathbb{E}|\boldsymbol{y}^\top \boldsymbol{z}_j|^2} = O\left( \frac{1}{\eta^5 p^{3/2}} \right).
\end{aligned}$$

This, in conjunction with (35), establishes **Step 1**.

According to the Borel-Cantelli lemma, it suffices to prove that for any $\varepsilon > 0$, the probability $\mathbb{P}(|\boldsymbol{u}^\top \boldsymbol{G}(z)\boldsymbol{\psi} - \mathbb{E}\boldsymbol{u}^\top \boldsymbol{G}(z)\boldsymbol{\psi}| > \varepsilon) = O(p^{-1-c})$ to establish **Step 2**, where $c$ is a positive constant. Let $\mathbb{E}_0$ denote expectation and $\mathbb{E}_j$ denote conditional expectation with respect to the $\sigma$-field generated by $\boldsymbol{x}_1, ..., \boldsymbol{x}_j$. Using the Schur complement formula again, we write

$$\begin{aligned}
\boldsymbol{u}^\top \boldsymbol{G}(z)\boldsymbol{\psi} - \mathbb{E}\boldsymbol{u}^\top \boldsymbol{G}(z)\boldsymbol{\psi} &= \sum_{j=1}^{n} (\mathbb{E}_j - \mathbb{E}_{j-1}) \boldsymbol{u}^\top \boldsymbol{G}(z)\boldsymbol{\psi} \\
&= \sum_{j=1}^{n} (\mathbb{E}_j - \mathbb{E}_{j-1})[\boldsymbol{u}^\top \boldsymbol{G}(z)\boldsymbol{\psi} - \boldsymbol{u}^\top \hat{\boldsymbol{G}}_j(z)(\boldsymbol{\psi} - \boldsymbol{e}_j \psi_j)] \\
&= \sum_{j=1}^{n} (\mathbb{E}_j - \mathbb{E}_{j-1})[-u_j \beta_j \psi_j + u_j \beta_j \boldsymbol{x}_j^\top \boldsymbol{X}_j \boldsymbol{G}_j \boldsymbol{\psi}_j + \psi_j \beta_j \boldsymbol{u}_j^\top \boldsymbol{G}_j \boldsymbol{X}_j^\top \boldsymbol{x}_j + \beta_j \boldsymbol{u}_j^\top \boldsymbol{G}_j \boldsymbol{X}_j^\top \boldsymbol{x}_j \boldsymbol{x}_j^\top \boldsymbol{X}_j \boldsymbol{G}_j \boldsymbol{\psi}_j] \\
&= \sum_{j=1}^{n} (\mathbb{E}_j - \mathbb{E}_{j-1})[\gamma_{j1} + \gamma_{j2} + \gamma_{j3} + \gamma_{j4}],
\end{aligned}$$

where $\boldsymbol{\psi}_j = [\psi_1, ..., \psi_{j-1}, \psi_{j+1}, ..., \psi_n]$. According to Lemma A.7, we have

$$\begin{aligned}
\mathbb{E}\left| \sum_{j=1}^{n} (\mathbb{E}_j - \mathbb{E}_{j-1})\gamma_{j1} \right|^4 &\le c\mathbb{E}\left( \sum_{j=1}^{n} \mathbb{E}|\gamma_{j1}|^2 \right)^2 + c\sum_{j=1}^{n} \mathbb{E}|\gamma_{j1}|^4 \\
&\le \frac{c}{\eta^4} \left( \sum_{j=1}^{n} u_j^2 \mathbb{E}\psi_j^2 \right)^2 + \frac{c}{\eta^4} \sum_{j=1}^{n} u_j^4 \mathbb{E}\psi_j^4 \\
&= O\left( \frac{1}{\eta^4 p^2} \right).
\end{aligned}$$

Similarly, we obtain

$$\mathbb{E}\left| \sum_{j=1}^{n} (\mathbb{E}_j - \mathbb{E}_{j-1})\gamma_{j2} \right|^4 \le \frac{c}{\eta^4} \left( \sum_{j=1}^{n} u_j^2 \mathbb{E}|\boldsymbol{x}_j^\top \boldsymbol{X}_j \boldsymbol{G}_j \boldsymbol{\psi}_j|^2 \right)^2 + \frac{c}{\eta^4} \sum_{j=1}^{n} u_j^4 \mathbb{E}|\boldsymbol{x}_j^\top \boldsymbol{X}_j \boldsymbol{G}_j \boldsymbol{\psi}_j|^4.$$

Denote $\boldsymbol{r}_j^\top = \boldsymbol{\psi}_j^\top \boldsymbol{G}_j(z) \boldsymbol{X}_j^\top$. It is apparent that

$$\|\boldsymbol{r}_j\|^2 = \boldsymbol{r}_j^\top \bar{\boldsymbol{r}}_j = \boldsymbol{\psi}_j^\top \boldsymbol{G}_j(z) \boldsymbol{X}_j^\top \boldsymbol{X}_j \boldsymbol{G}_j(\bar{z}) \boldsymbol{\psi}_j.$$

According to Lemma A.1 and the identity

$$\boldsymbol{G}_j(z) \boldsymbol{X}_j^\top \boldsymbol{X}_j \boldsymbol{G}_j(\bar{z}) = \boldsymbol{G}_j(\bar{z}) + z\boldsymbol{G}_j(z)\boldsymbol{G}_j(\bar{z}) = \boldsymbol{G}_j(\bar{z}) + z\frac{\boldsymbol{G}_j(z) - \boldsymbol{G}_j(\bar{z})}{z - \bar{z}},$$

we get for $m \geq 1$,

$$
\begin{aligned}
\mathbb{E}\|\boldsymbol{r}_j\|^{2m} &\leq c\mathbb{E}\left(\boldsymbol{\psi}_j^{\mathsf{T}}\boldsymbol{G}_j(z)\boldsymbol{X}_j^{\mathsf{T}}\boldsymbol{X}_j\boldsymbol{G}_j(\bar{z})\boldsymbol{\psi}_j - \mathbb{E}\boldsymbol{\psi}_j^{\mathsf{T}}\boldsymbol{G}_j(z)\boldsymbol{X}_j^{\mathsf{T}}\boldsymbol{X}_j\boldsymbol{G}_j(\bar{z})\boldsymbol{\psi}_j\right)^m + c\left(\mathbb{E}\boldsymbol{\psi}_j^{\mathsf{T}}\boldsymbol{G}_j(z)\boldsymbol{X}_j^{\mathsf{T}}\boldsymbol{X}_j\boldsymbol{G}_j(\bar{z})\boldsymbol{\psi}_j\right)^m \\
&\leq C\mathbb{E}\left(\left|\boldsymbol{\psi}_j^{\mathsf{T}}\boldsymbol{G}_j(z)\boldsymbol{\psi}_j - \mathbb{E}\boldsymbol{\psi}_j^{\mathsf{T}}\boldsymbol{G}_j(z)\boldsymbol{\psi}_j\right|^m + \frac{1}{\eta^m}\left|\boldsymbol{\psi}_j^{\mathsf{T}}[\boldsymbol{G}_j(z) - \boldsymbol{G}_j(\bar{z})]\boldsymbol{\psi}_j - \mathbb{E}\boldsymbol{\psi}_j^{\mathsf{T}}[\boldsymbol{G}_j(z) - \boldsymbol{G}_j(\bar{z})]\boldsymbol{\psi}_j\right|^m\right) \\
&\quad + C\left|\mathbb{E}\boldsymbol{\psi}_j^{\mathsf{T}}\frac{\boldsymbol{G}_j(z) - \boldsymbol{G}_j(\bar{z})}{z - \bar{z}}\boldsymbol{\psi}_j\right|^m + C|\mathbb{E}\boldsymbol{\psi}_j^{\mathsf{T}}\boldsymbol{G}_j(z)\boldsymbol{\psi}_j|^m \\
&= O(1),
\end{aligned}
$$

where we use Lemma A.1 and the observations that

$$
\mathbb{E}\boldsymbol{\psi}_j^{\mathsf{T}}\frac{\boldsymbol{G}_j(z) - \boldsymbol{G}_j(\bar{z})}{z - \bar{z}}\boldsymbol{\psi}_j = \frac{\sum_{a=1}^{K} c_a[m_a(z) - m_a(\bar{z})]}{p[z - \bar{z}]}\mathrm{tr}\boldsymbol{C}_a^2 + o(1),
$$

$$
c_0\frac{m_a(z) - m_a(\bar{z})}{z - \bar{z}} = \int \frac{1}{|\lambda - z|^2}\mathrm{d}\nu_a(\lambda) \geq \frac{1}{\mathrm{dist}^2(\mathcal{S}, z)}.
$$

Following a similar argument as in (36), we have for $m \geq 1$,

$$
\mathbb{E}|\boldsymbol{x}_j^{\mathsf{T}}\boldsymbol{X}_j\boldsymbol{G}_j\boldsymbol{\psi}_j|^{2m} = \mathbb{E}|\boldsymbol{r}_j^{\mathsf{T}}\boldsymbol{x}_j|^{2m} = \mathbb{E}\mathbb{E}[|\boldsymbol{r}_j^{\mathsf{T}}\boldsymbol{x}_j|^{2m} \mid \boldsymbol{X}_j] \leq c\mathbb{E}\frac{\|\boldsymbol{r}_j\|^m + \|\boldsymbol{r}_j\|^{2m}}{p^m} = O\left(\frac{1}{p^m}\right).
$$

Therefore, we have $\sum_{j=1}^{n}(\mathbb{E}_j - \mathbb{E}_{j-1})\gamma_{j2} \to 0$ a.s. by showing that

$$
\mathbb{E}\left|\sum_{j=1}^{n}(\mathbb{E}_j - \mathbb{E}_{j-1})\gamma_{j2}\right|^4 \leq \frac{c}{\eta^4}\left[\left(\sum_{j=1}^{n}\frac{u_j^2}{p}\right)^2 + c\sum_{j=1}^{n}\frac{u_j^4}{p^2}\right] = O\left(\frac{1}{\eta^4 p^2}\right).
$$

Similarly, for $m \geq 1$, it can be obtained that

$$
\mathbb{E}|\gamma_{j3}|^{2m} = \mathbb{E}|\psi_j\beta_j\boldsymbol{u}_j^{\mathsf{T}}\boldsymbol{G}_j\boldsymbol{X}_j^{\mathsf{T}}\boldsymbol{x}_j|^{2m} \leq \frac{1}{\eta^{2m}}\sqrt{\mathbb{E}\psi_j^{4m}\mathbb{E}|\boldsymbol{y}_j^{\mathsf{T}}\boldsymbol{x}_j|^{4m}} = O\left(\frac{1}{\eta^{6m}p^{2m}}\right) \tag{37}
$$

and

$$
\mathbb{E}|\gamma_{j4}|^{2m} = \mathbb{E}|\beta_j\boldsymbol{u}_j^{\mathsf{T}}\boldsymbol{G}_j\boldsymbol{X}_j^{\mathsf{T}}\boldsymbol{x}_j\boldsymbol{x}_j^{\mathsf{T}}\boldsymbol{X}_j\boldsymbol{G}_j\boldsymbol{\psi}_j|^{2m} \leq \frac{1}{\eta^{2m}}\sqrt{\mathbb{E}|\boldsymbol{y}_j^{\mathsf{T}}\boldsymbol{x}_j|^{4m}|\boldsymbol{r}_j^{\mathsf{T}}\boldsymbol{x}_j|^{4m}} = O\left(\frac{1}{\eta^{10m}p^{2m}}\right). \tag{38}
$$

One can conclude by Lemma A.7 that

$$
\left|\sum_{j=1}^{n}(\mathbb{E}_j - \mathbb{E}_{j-1})\gamma_{j3}\right|^4 \leq c\left(\sum_{j=1}^{n}\mathbb{E}|\gamma_{j3}|^2\right)^2 + c\sum_{j=1}^{n}\mathbb{E}|\gamma_{j3}|^4 = O\left(\frac{1}{p^2\eta^{12}}\right)
$$

and

$$
\left|\sum_{j=1}^{n}(\mathbb{E}_j - \mathbb{E}_{j-1})\gamma_{j4}\right|^4 \leq c\left(\sum_{j=1}^{n}\mathbb{E}|\gamma_{j4}|^2\right)^2 + c\sum_{j=1}^{n}\mathbb{E}|\gamma_{j4}|^4 = O\left(\frac{1}{p^2\eta^{20}}\right).
$$

Combining the results above, one can conclude that

$$
\begin{aligned}
\mathbb{P}\left(\left|\sum_{j=1}^{n}(\mathbb{E}_j - \mathbb{E}_{j-1})\left[\gamma_{j1} + \gamma_{j2} + \gamma_{j3} + \gamma_{j4}\right]\right| > \varepsilon\right) &\leq \varepsilon^{-4}\mathbb{E}\left|\sum_{j=1}^{n}(\mathbb{E}_j - \mathbb{E}_{j-1})\left[\gamma_{j1} + \gamma_{j2} + \gamma_{j3} + \gamma_{j4}\right]\right|^4 \\
&\leq C\varepsilon^{-4}\sum_{k=1}^{4}\mathbb{E}\left|\sum_{j=1}^{n}(\mathbb{E}_j - \mathbb{E}_{j-1})\gamma_{jk}\right|^4 \\
&= O\left(\frac{1}{\eta^{20}p^2}\right).
\end{aligned}
$$

Hence, the proof of **Step** 2 is completed. Moreover, for any $z$ with $\Re z < 0$, we can obtain the result by replacing $\eta$ by $|\Re z|$ in the proof. Although we assume $z$ is fixed, the same conclusion also holds for $\eta = \eta_n$ converging to 0 sufficiently slowly such as $\eta_n = (\log n)^{-1}$, when $\Re z$ is bounded away from $\mathcal{S} \cup \{0\}$. We notice that $\boldsymbol{u}^\mathsf{T} \boldsymbol{G}(z) \boldsymbol{\psi}$ is an analytic function on any open set $O \subset \mathbb{C}$ if $\mathrm{dist}(\mathrm{Spec}(\boldsymbol{X}^\mathsf{T} \boldsymbol{X}), O)$ is away from 0. Recall that $\mathrm{dist}(\mathrm{Spec}(\boldsymbol{X}^\mathsf{T} \boldsymbol{X}), \mathcal{S} \cup \{0\}) \to 0$ almost surely, for $z \in O$ with $0 \leq |\Im z| \leq \eta_n$ and $\mathrm{dist}(\mathcal{S} \cup \{0\}, \Re z) > c$ for some positive constant $c$. It is clear that

$$|\boldsymbol{u}^\mathsf{T} \boldsymbol{G}(z) \boldsymbol{\psi}| \leq |\boldsymbol{u}^\mathsf{T} \boldsymbol{G}(z) \boldsymbol{\psi} - \boldsymbol{u}^\mathsf{T} \boldsymbol{G}(z + \mathrm{i}\eta_n) \boldsymbol{\psi}| + |\boldsymbol{u}^\mathsf{T} \boldsymbol{G}(z + \mathrm{i}\eta_n) \boldsymbol{\psi}| \to 0 \quad a.s.$$

Therefore, we conclude **Step 3** and complete the proof of Lemma A.2.

### A.5. Auxiliary lemmas

This section contains several established results from previous work which are needed in our proof.

**Lemma A.3.** *(Theorem A.43 in Bai & Silverstein (2010)). Let $\boldsymbol{A}$ and $\boldsymbol{B}$ be two $n \times n$ Hermitian matrices, and let $F^{\boldsymbol{A}}$ and $F^{\boldsymbol{B}}$ denote their respective empirical spectral distributions. Then*

$$\|F^{\boldsymbol{A}} - F^{\boldsymbol{B}}\| \leq \frac{1}{n} \mathrm{rank}(\boldsymbol{A} - \boldsymbol{B}).$$

**Lemma A.4.** *(Theorem A.46 in Bai & Silverstein (2010)). Let $\boldsymbol{A}$ and $\boldsymbol{B}$ be two Hermitian matrices. Then*

$$\max_i |\lambda_i(\boldsymbol{A}) - \lambda_i(\boldsymbol{B})| \leq \|\boldsymbol{A} - \boldsymbol{B}\|.$$

**Lemma A.5.** *Let $f_1, f_2, \ldots$ be analytic on D, a connected open set of $\mathbb{C}$, satisfying $|f_n(z)| \leq M$ for every $n$ and $z$ in D, and $f_n(z)$ converges, as $n \to \infty$ for each $z$ in a subset of D having a limit point in D. Then there exists a function $f$, analytic in D for which $f_n(z) \to f(z)$ and $f_n'(z) \to f'(z)$ for all $z \in D$. Moreover, on any set bounded by a contour interior to D the convergence is uniform and $f_n(z)$ is uniformly bounded by $2M/\epsilon$, where $\epsilon$ is the distance between the contour and the boundary of D.*

The proof of Lemma A.5 can be found in Bai & Silverstein (2004).

**Lemma A.6.** *(Lemma 2.2 in Bai & Silverstein (2004)). For $\boldsymbol{z} = [z_1, \ldots, z_n]^\mathsf{T}$ with i.i.d. standardized random entries, $\boldsymbol{C}$ is an $n \times n$ matrix, we have, for any $m \geq 2$,*

$$\mathbb{E}|\boldsymbol{z}^\mathsf{T} \boldsymbol{C} \boldsymbol{z} - \mathrm{tr}\boldsymbol{C}|^m \leq C[(\mathbb{E}|z_1|^4 \mathrm{tr}\boldsymbol{C}\boldsymbol{C}^*)^{\frac{m}{2}} + \mathbb{E}|z_1|^{2m} \mathrm{tr}(\boldsymbol{C}\boldsymbol{C}^*)^{\frac{m}{2}}].$$

**Lemma A.7.** *(Burkholder's inequality). Let $\{x_k\}$ be a complex martingale difference sequence with respect to the increasing $\sigma$-field $\{\mathcal{F}_k\}$. Then for any $m \geq 2$,*

$$\mathbb{E}\left|\sum x_k\right|^m \leq c_m \mathbb{E}\left(\sum \mathbb{E}(|x_k|^2 | \mathcal{F}_k)\right)^{\frac{m}{2}} + c_m \sum \mathbb{E}|x_k|^m.$$

**Lemma A.8.** *(Sylverster's determinant identity). If $\boldsymbol{A}$ and $\boldsymbol{B}$ are matrices of sizes $m \times n$ and $n \times m$, then*

$$\det(\boldsymbol{I}_m + \boldsymbol{A}\boldsymbol{B}) = \det(\boldsymbol{I}_n + \boldsymbol{B}\boldsymbol{A}).$$

