# OpenReview forum: "Eigen Analysis of Conjugate Kernel and Neural Tangent Kernel"
_ICML.cc/2025/Conference — ICML 2025 poster_

### Official Review · Reviewer_u6qk · 2025-03-12

**Overall Recommendation:** 4

**Summary:**

This paper analyzes the spectral properties of deep feedforward neural networks with random weights, focusing on Gaussian mixture model inputs. It rigorously examines isolated eigenvalues in the conjugate and neural tangent kernels, showing how they capture group features and evolve through hidden layers, influenced by covariance differences and activation functions.

**Claims And Evidence:**

Yes.

**Essential References Not Discussed:**

NO.

**Experimental Designs Or Analyses:**

Yes.

**Methods And Evaluation Criteria:**

Yes.

**Other Comments Or Suggestions:**

No

**Other Strengths And Weaknesses:**

Strengths:

1 This paper studies the Stieltjes transforms of CKs and NTKs using the deterministic equivalents provided by previous work (Gu et al., 2024). The work is highly thorough, and the theoretical contribution is solid.

2  The precise characterization of the isolated eigenvalues is very interesting.

Weaknesses:

1 Regarding technical novelty, the spectra of CKs and NTKs with linear width have been well studied by (Fan & Wang, 2020). This paper, however, explores the case of extremely large width. The prior width setting is more practical and extends beyond lazy learning, with the development of a new theoretical framework. Moreover, the deterministic equivalents of CKs and NTKs were already provided in previous work (Gu et al., 2024), and this paper examines the same NN model. Standard random matrix tools can easily derive results for the Stieltjes transforms, isolated eigenvalues, and eigenvectors. The technique used here is standard. If there is any new technical contribution I missed, please let me know.

2 While this paper provides very complete results, the relevance between the theoretical findings and learning tasks is weak. Particularly for readers unfamiliar with RMT, the theoretical results may not offer novel insights.  For example,  (Gu et al., 2024) applies their theoretical results to design new compression methods. For this paper, my suggestion is to discuss  the impact of the isolated eigenvalues on training dynamics or generalization performance. It could provide further insight.

Overall, I am positive about this paper, but I encourage the authors to further explore the theoretical results in the context of learning.

**Questions For Authors:**

No

**Relation To Broader Scientific Literature:**

The most relevant work is (Gu et al., 2024), which serves as a complementary study to this paper.

**Theoretical Claims:**

Yes. I have checked all of the proofs.

---

> ### Author Rebuttal · Authors · 2025-03-31
>
> We sincerely thank the reviewer for their kind and encouraging feedback on our paper: "The work is highly thorough, and the theoretical contribution is solid. The precise characterization of the isolated eigenvalues is very interesting". This positive recognition motivates us greatly. Below, we provide detailed point-by-point responses (marked with $\Huge{\cdot}$) to address all identified weaknesses. The statements of weaknesses have been condensed to comply with character limitations.
> > Regarding technical novelty,..., let me know.
> * Fan and Wang (2020) proved the limiting spectral distributions of both CK and NTK under the linear width assumption. Subsequently, Wang et al. (2024) established the asymptotic behavior of spiked eigenvalues and eigenvectors of CK. In this paper, while our theoretical results are derived using the standard random matrix framework based on the pioneering work of Gu et al. (2024), they reveal intricate phenomena involving the isolated eigenvalues and eigenspaces of neural networks. Specifically, as outlined in Section 3.2, the first-order limits of entries in isolated eigenvectors of both CK and NTK may or may not encode group features. To clarify the differences between Wang et al. (2024) and our study, we provide the following comparisons.
> 1. We allow for different population covariance matrices $C_a$ in the GMM data, without relying on the $\tau_n-$orthonormal assumption introduced in Wang et al. (2024). Specifically, the $\tau_n-$orthonormal assumption requires that the difference between $tr(C_a)$’s be of order $o(p^{2/3})$, whereas our framework only requires this difference to be $O(p)$.
> 2. We do not require $X^TX$ to contain isolated eigenvalues, as specified in Assumption 2 of Wang et al. (2024). Instead, the isolated eigenvalues of CK and NTK can arise from the group membership structure of the input data, i.e., through the terms $VA_{\ell}V^T$ and $VB_{\ell}V^T$ in our equations (5) and (8).
> 3. They assume isolated eigenvalues are simple, focusing on single eigenvectors in their main theorems. In contrast, our work accommodates isolated eigenvalues with multiplicities greater than one and analyzes the corresponding eigenspaces.
> 4. The two studies adopt different assumptions regarding the activation functions. Their work employs weaker conditions on the moments of the derivatives, whereas ours imposes weaker constraints on the expectations of the first and second derivatives.
> 5. Given NTK's ability to approximate the generalization and dynamics of neural networks (NNs) on real data, our work explores the properties of NTK which are not addressed in Wang et al. (2024).
>
> Additionally, we conducted experiments on the MNIST dataset under conditions that satisfy the linear width assumption. The empirical results support the existence of distinct isolated eigenvectors as revealed by our theoretical findings. Please click the following link to view the results: <https://anonymous.4open.science/r/eig5492/README.md>.
>
> > While this paper provides very complete results, ..., provide further insight.
> * By analyzing NTK in ultra-wide NNs, our theoretical findings provide insights into the dynamics of training in such architectures. Let $\boldsymbol{y}$ denote the label vector of the data. When the loss function is defined as $L(t)=\|\|\boldsymbol{y}-\hat{\boldsymbol{y}}(t) \|\|^2$, where $\hat{\boldsymbol{y}}(t)$ represents the prediction at time $t$, the time evolution of the residual is approximately given by the ODE: $$\frac{d}{dt} \hat{\boldsymbol{y}} (t)= \boldsymbol{K}_{\text{NTK}} \big(\boldsymbol{y} -\hat{\boldsymbol{y}} (t)).$$ This indicates that the NTK captures the dynamics of the training process in ultra-wide NNs. We draw the following statistical implications:
> 1. Based on our theoretical results from Theorem 3.7 to Remark 3.9, the first-order limits of entries in the isolated eigenvectors may or may not contain group features.
> 2. When the eigenspace associated with the largest isolated eigenvalue contains group features, NNs tend to prioritize learning this information. Conversely, if this eigenspace lacks group features, NNs instead prioritize learning irrelevant information, diverting attention away from effective group features.
> 3. For non-isolated eigenspaces, we conjecture that their first-order limits contain significantly fewer group features, as observed in classical random matrices like the sample covariance matrix, where the entries of non-spiked eigenvectors are delocalized under mild conditions. Consequently, after effectively learning the group features captured by eigenspaces corresponding to isolated eigenvalues, NNs primarily shift their focus to learning from eigenspaces that lack first-order group features. In GMM setting, this implies that NNs begin leveraging more noise to learn $\boldsymbol{y}$, resulting in overfitting.
> This potentially provides a perspective based on NTK to understand the phenomenon of overfitting.

---

> > ### Comment · Reviewer_u6qk · 2025-04-02
> >
> > I would like to thank the authors for their detailed rebuttal. While I do not think that the inherent limitations of this paper can be fully addressed in the rebuttal period, I appreciate the authors' efforts. I will revise my score to 4 as an encouragement.

---

> > > ### Author Response · Authors · 2025-04-03
> > >
> > > We sincerely thank the reviewer for his/her kind and encouraging support.

---

### Official Review · Reviewer_tzoT · 2025-03-12

**Overall Recommendation:** 3

**Summary:**

The paper explores the eigenvalue spectrum of the conjugate kernel (CK) and the neural tangent kernel (NTK) with random weights. The authors demonstrate the existence of isolated eigenvalues and present a theoretical approach to identifying where they lie and their possible impact on the model.

## Update after rebuttal

After the rebuttal period, I felt the authors expanded on their key points in a suitable way based on the criticism. However, I did not feel that they clearly outlined the important points and conclusion of the paper and how it builds on the current state of the art. I feel that with additional experiments and discussion related to their conclusions, the paper would be a stronger contribution. However, the work was well performed and so I am happy to raise my score by one point.

**Claims And Evidence:**

There are very few claims made in the paper rather the authors highlight specific results and theoretical insights. To this end, there is evidence to support their claims.

**Essential References Not Discussed:**

More citations could be included regarding similar work studying the spectra of the NTK. However, it would not support the experiments or results of the manuscript.

**Experimental Designs Or Analyses:**

The authors do not present a variety of experiments or testable hypotheses.

**Methods And Evaluation Criteria:**

Very few benchmarks and datasets are used to demonstrate the paper's main results. It focuses far more on theoretical advances.

**Other Comments Or Suggestions:**

I would suggest that the authors apply their analysis to more common network architectures and try to connect their results to understandable results in machine learning.

**Other Strengths And Weaknesses:**

The paper doesn't explain why these results are novel or relevant. The results appear well validated and perhaps interesting. However, how can a machine learning practitioner, theoretical or otherwise, use these results to learn more about their networks/activation functions or training?

**Questions For Authors:**

1. What role do the authors expect the isolated eigenvalues play in the networks during training?
2. How do you expect the results to change when applied to more architectures?

**Relation To Broader Scientific Literature:**

There are many references made to alternative research performed on the spectra of the NTK or CK matrices. The authors do not relate their work too extensively to this literature, mostly introducing their own ideas.

**Theoretical Claims:**

I did not see any obvious theoretical issues but did not check all equations closely.

---

> ### Author Rebuttal · Authors · 2025-04-01
>
> We sincerely thank the reviewer for the constructive questions. In the following, we provide point-by-point responses (marked with $\Huge{\cdot}$).
> > The paper doesn't explain why these results are novel or relevant. The results appear well validated and perhaps interesting. However, how can a machine learning practitioner, theoretical or otherwise, use these results to learn more about their networks/activation functions or training?
> * We thank the reviewer for this insightful comment. Below, we clarify the novelty and practical implications of our findings.
> 1. Theoretical Insights into NTK and CK:
>    - Our theoretical results in Section 3 reveal that the first-order limits of entries in isolated eigenvectors, derived from both NTK and CK, may or may not encode group features.
>    - To empirically verify this, we conducted an experiment using the MNIST dataset. A histogram showing the spectrum of the third layer's CK matrix and line plots of the eigenvectors for the top 5 eigenvalues can be found at the following link: <https://anonymous.4open.science/r/eig5492/README.md>.
> The spectrum shows four isolated eigenvalues, with the eigenvector corresponding to the largest eigenvalue being informative, while the other three appear non-informative. These observations align with our theoretical findings from Theorem 3.4 to Remark 3.6.
>
> 2. Training Dynamics in Ultra-Wide Neural Networks (NNs):
>    - By analyzing NTK in ultra-wide NNs, we uncover deeper implications regarding the dynamics of training in such architectures. Let $\boldsymbol{y}$ represent the label vector of the data. When the loss function $L(t)=\|\|\boldsymbol{y}-\hat{\boldsymbol{y}}(t) \|\|^2 $, where $\hat{\boldsymbol{y}}(t)$ denotes the prediction at time $t$, the time evolution of the residual is approximately given by the ODE: $\frac{d}{dt} \hat{\boldsymbol{y}} (t)= \boldsymbol{K}_{\text{NTK}}\big(\boldsymbol{y} -\hat{\boldsymbol{y}} (t)).$ This indicates that the NTK captures the dynamics of the training process in ultra-wide NNs.
>    - When the eigenspace associated with the largest isolated eigenvalue contains group features, NNs tend to prioritize learning this information. Conversely, if this eigenspace lacks group features, NNs instead prioritize learning irrelevant information, diverting attention away from effective group features.
>    - For non-isolated eigenspaces, we conjecture that their first-order limits contain significantly fewer group features, as observed in classical random matrices like the sample covariance matrix, where the entries of non-spiked eigenvectors are delocalized under mild conditions. Consequently, after effectively learning the group features captured by eigenspaces corresponding to isolated eigenvalues, NNs primarily shift their focus to learning from eigenspaces that lack first-order group features. In GMM, this implies that NNs begin leveraging more noise to learn $\boldsymbol{y}$, resulting in overfitting. This potentially provides a perspective based on NTK to understand the phenomenon of overfitting.
>
> 3. Impact of Activation Functions on Group Features Encoding:
>    - As demonstrated in Figure 4 of our paper, different activation functions yield distinct behaviors in the eigenvalues and eigenvectors. A carefully selected activation function (e.g., ReLU in Figure 4(b) and 4(d)) can effectively encode group features within NNs, facilitating the integration of group features from the initial stages.
>    - These findings suggest that careful selection of activation functions can significantly enhance a NN's ability to capture and represent group features.
>
> > What role do the authors expect the isolated eigenvalues play in the networks during training?
> * Please refer to our response to the previous question, specifically points 1 and 2, which address the role of isolated eigenvalues in networks during training.
>
> > How do you expect the results to change when applied to more architectures?
> * We believe that isolated eigenvalues and eigenspaces are likely to exist across various NN architectures (e.g., fully connected neural networks (FCNs) with dropout), provided the input data exhibits certain classification characteristics. Investigating these isolated eigenvalues and eigenspaces in different NN structures remains a largely unexplored problem. However, under current theoretical frameworks, it is challenging to directly prove the existence of isolated eigenvalues and eigenspaces in architectures different from FCN. For instance, whether the spectral equivalence described in Lemma 2.4 and Lemma 2.5 holds for other NN architectures remains unknown. We aim to address these related challenges in our future work.
>
> Furthermore, in the revision we will include more references on the spectra of NTK and CK matrices, such as Pennington and Worah (2017, NIPS), Yang and Salman (2019, arxiv), Hron et al. (2020, ICML), Murray et al.(2022, arxiv), Wang et al. (2023, NIPS), and Belfer (2024, JMLR).

---

> > ### Comment · Reviewer_tzoT · 2025-04-02
> >
> > I thank the authors for the detailed explanation of the core results. I still feel that the message is convoluted at times and if it were presented clearly, the authors would benefit greatly. I have a few comments regarding the rebuttal.
> >
> > * "may or may not encode group features" is quite ambiguous. What should I take from that sentence?
> > * From my understanding, it is widely known that eigenvectors of dominant eigenvalues are important in training, but this was also extended to additional vectors in understanding evolution, e.g., in https://arxiv.org/abs/1812.04754. But to argue this definitively, the authors would need to show that it holds on more datasets. It could simply be true that this only occurs on the dataset you studied.
> > * Why is it surprising that the NTK captures training dynamics in ultra-wide networks? The matrix describes the state of a network given its data and if this state changes, so too will the NTK. If the network is very wide, this change becomes very slow / small which corresponds to a small change in the NTK. Eventually, this would be seen a small enough change for the difference in NTK states to be descriptive.
> > * Activation functions helping to discern features in data is of interest, but I would like to be convinced that this analysis goes further than other works.
> > * Perhaps something that would help with explaining the results here. If the authors wanted to fit a problem "perfectly", ideal data distribution and network, what would the spectrum need to look like?

---

> > > ### Author Response · Authors · 2025-04-06
> > >
> > > We are truly grateful to the reviewer for the prompt feedback. Below, we provide point-by-point responses (once again marked with $\Huge{\cdot}$).
> > > >  "may or may not encode group features" is quite ambiguous. What should I take from that sentence?
> > > * By the expression "may or may not encode group features", we mean that the eigenspace corresponding to isolated eigenvalues can either be informative or non-informative, as defined in Definition 3.3 of our paper. For your convenience, we attach the definition here:
> > >   > Definition 3.3. We say the eigenspace defined in (20)  is informative if there is a nonzero matrix $\mathbf{A}(\boldsymbol{J})$ depending on $\boldsymbol{J}$ such that $$\|\|\frac{1}{p}\boldsymbol{J}\^T\widehat{\boldsymbol{\Pi}}\_{\rho}\boldsymbol{J}-\mathbf{A}(\boldsymbol{J})\|\|=o_{a.s.}(1).$$ Otherwise if $\mathbf{A}(\boldsymbol{J})=0$, then the eigenspace is non-informative.
> > >   - Recall that $\boldsymbol{J}=[\boldsymbol{j}\_1,...,\boldsymbol{j}\_K]\in\mathbb{R}\^{n\times K}$, where $\boldsymbol{j}\_a=\\{\delta\_{{x_i}\in\mathcal{C}\_a}\\}\_{i=1}^n$ encodes the group membership information in GMM. The matrix $\mathbf{A}(\boldsymbol{J})$ reflects information about the projection of $\boldsymbol{J}$ onto the eigenspaces corresponding to isolated eigenvalues. A nonzero value indicates the eigenspace contains membership information, i.e., it encodes group features. In constrast, a zero value implies that such eigenspace is orthogonal to $\boldsymbol{J},$ rendering it non-informative and not encoding group features.
> > >   - Our theoretical results in Section 3.2 demonstrate that the informativeness of an eigenspace corresponding to an isolated eigenvalue is jointly determined by the covariance structure of the input data and the choice of activation function. Specifically, this is established in Theorem 3.4—Remark 3.6 for CK, and Theorem 3.7—Remark 3.9 for NTK.
> > >   - Furthermore, previous experiments on MNIST (<https://anonymous.4open.science/r/eig5492/README.md>) align with our theoretical findings. The spectrum exhibits four isolated eigenvalues, with the eigenvector corresponding to the largest eigenvalue being informative (clearly distinguishing two classes), while the remaining three seem to be non-informative.
> > >
> > > > From my understanding, ... studied.
> > > * Following your suggestion, we conducted additional experiments using another dataset, CIFAR-10.  Similar to the MNIST dataset, we present the spectrum histogram and eigenvector plots; see Figures 1-3 at the new link: <https://anonymous.4open.science/r/C5492-FC8C/README.md>. The spectrum shows nine isolated eigenvalues, with the four largest eigenvectors being informative and the remaining five non-informative—similar to observations from MNIST.
> > >
> > > > Why is it surprising that the NTK ... descriptive.
> > > * We appreciate your insightful comments and fully agree with the points you've raised. To clarify, what surprises us is not the observation that the NTK captures training dynamics in ultra-wide NNs. Rather, what we find interesting—and what our study highlights through explicit eigen analysis of the NTK—is that even within the low-dimensional top subspace, the eigenspaces corresponding to isolated eigenvalues can differ significantly in informativeness. While some isolated eigenspaces are indeed informative, encoding meaningful group features, others may be non-informative. This nuanced finding, as detailed in our response to your first comment, offers us a deeper understanding of the NTK's role in training dynamics.
> > >
> > > > Activation functions ... works.
> > > * Regarding isolated eigenvalues and eigenvectors in CK and NTK, the most recent study we are aware of is Wang et al. (2024) on CK. In our response to Reviewer Hb59's first comment, we provided a detailed comparison with their work. Specifically, in terms of activation functions, their work employs weaker conditions on the order of derivatives, while ours imposes weaker constraints on the expectations of the first and second derivatives. Moreover, we allow the use of different activation functions across different layers, providing enhanced flexibility in activation function choices for practical applications.
> > >
> > > > Perhaps something ... look like?
> > > * If we understand the comment correctly, it inquires about the structure of the CK or NTK spectrum in the ideal case where the data, network, and all assumptions in our paper are satisfied. We believe that typical spectra of CK and NTK exhibit the following structure: the non-zero eigenvalues ($\lambda_n\le\ldots\le\lambda_1$), except for the isolated ones, are clustered within one or more intervals. Within each interval, the eigenvalues are tightly grouped $(\lambda_i-\lambda_{i+1}=o_p(1))$. In contrast, the isolated eigenvalues are distinctly separated from these intervals as well as from zero.  For reference, we conducted an experiment to visualize this structure; please see Figure 4 at the new link: <https://anonymous.4open.science/r/C5492-FC8C/README.md>.

---

### Official Review · Reviewer_xuzj · 2025-03-14

**Overall Recommendation:** 4

**Summary:**

This paper analyzes the eigenvalues and eigenvectors of the Conjugate Kernel (CK) and Neural Tangent Kernel (NTK) for deep feedforward networks with random weights, in a high-dimensional setting where the input data come from a Gaussian Mixture Model (GMM). The authors show that, under certain assumptions, “spiked” or isolated eigenvalues of the CK and NTK can emerge outside the main (bulk) support of the limiting spectral distribution. They then precisely characterize the limiting positions of these outlier eigenvalues and analyze the asymptotic structure of the eigenvectors (or eigenspaces). The main conceptual point is that these isolated eigenvalues reflect group (or cluster) structure in the data—i.e. the eigenvectors encode the mixture components. Intuitively, the paper demonstrates how features of the data propagate layer by layer in a randomly initialized deep network, indicating how certain group structure is captured in these outlier modes. Overall, it provides theoretical insight into how CK/NTK spectra depart from their bulk distribution in the presence of class- or cluster-relevant structure.


## update after rebuttal

I am happy with the rebuttal responses and will maintain my originally high score 4 (Accept). **Important caveat**  I do agree with reviewer `Hb59` that a proper discussion of Wang et. al. 2024 is highly warranted given the high degree of similarity.   If the paper is accepted, Authors should be super clear about what is new or different in their results compared to the existing results.

**Claims And Evidence:**

While this is a theoretical contribution and the support of claims is primarily theoretical, here are limited experiments that support the key claims:

- Main Claim 1: “Certain eigenvalues of the CK/NTK can lie outside the bulk of the spectral measure in high-dimensional regimes.” Evidence: The authors present formal statements (Theorem 3.1 & 3.2) they show simulations that show outliers numerically.
- Main Claim 2: “These outlier eigenvalues correspond to group or cluster structure in the data.”(Theorem 3.4 & Corollary 3.5).
Evidence:  They provide numerical plots (eigenvectors strongly correlate with mixture class membership).
- Main Claim 3: “The position of these isolated eigenvalues and the structure of their eigenvectors can be characterized by expansions related to the data covariance and the choice of activation function.”
Evidence: The paper provides explicit formulas (involving terms α_ℓ,2, α_ℓ,3, etc.) for the CK and NTK spiked eigenvalues. The simulations with polynomial vs. ReLU activations illustrate how the location of outliers depends on activation.

**Essential References Not Discussed:**

the paper already cites much of the very relevant literature. But it might benefit from some other citation that are more broadly related:
- Neural Collapse (Papyan et al., 2020) for how class means or class structure leads to low-rank phenomena in feature space. This might give the authors a link between their results and the class structure that emerges naturally during training in neural collapse.
- Yang’s Tensor Programs (Yang, 2019–2021) might be relevant because it also addresses expansions of NTK under more general architectures. Although the paper’s setting is quite different, referencing it could highlight synergy between the two lines of theory.

**Experimental Designs Or Analyses:**

Overall, the experimental design matches the paper’s theoretical claims and is valid for a proof-of-concept demonstration:
- The experiments primarily involve plotting eigenvalue histograms and the top eigenvectors for various settings (e.g. polynomial vs. ReLU activation). This is sufficient for a theoretical paper.
- The dimensionalities (n, p) used are large enough (e.g. n=1200, p=600) to demonstrate the asymptotic effect, and the results show visually distinct outliers.

**Methods And Evaluation Criteria:**

the methods and evaluation approach (matching theoretical spectral predictions to empirical ones) are well-aligned with the paper’s theoretical goals:
- The authors’ primary method is an asymptotic random matrix theoretical derivation of eigenvalue limits. They compare theoretical predictions with histograms of empirical eigenvalues from random networks.
- The data are synthetic or follow a controlled GMM with different class covariances, which is reasonable for theoretical validation.

**Other Comments Or Suggestions:**

- As mentioned earlier, some discussion around neural collapse and feature learning literature would add value for some readers. Considering inputs to be clustered is a highly idealized and unrealistic initialization (as devil's advocate would ask: why would we need a neural network if the input is clustered?). Thus, making some connection to neural collapse would make the results more interesting for some readers.
- The paper celebrates its key finding as finding connection between outlier eigenvalues and the activation itself, via the recurrent equations defined over moments of the activation. But they do not explore differences of activations from the point of view of their theory. For example in Figure 4 a& b for  poly and ReLU activation look largely & qualitatively the same, which seems to underserve the key focus of this paper, which is uncovering role of activation on the magnitude and position of eigenvalue outliers. My suggestion is that authors try to draw some implications from their theory about the activations that are not currently known, and show that that empirically holds. Many future readers may find these practical insights to be be the key takeaways of this paper, and without them, they will be left with no takeaways.

**Other Strengths And Weaknesses:**

Strengths:
- Novelty: Detailed spiked eigenvalue analysis for CK/NTK in multi-layer random networks with GMM data is new and addresses a gap in the literature.
- Technical Rigor: The paper’s theorems are grounded in well-established RMT methods (deterministic equivalents, rank-1 updates).
- Clarity & Focus: the statements of main theorems and the structured proofs in the appendix are well written

Weaknesses:
- Data Model & Assumptions: Results rely on a GMM with somewhat idealized settings (independence, certain covariance scaling). If this is truly about initialization and the input is taken to be the real input, in reality, large-scale datasets do not always behave like mixtures of Gaussians. For example, the structure of the NTK and Conjugate kernels as revealed by lemma 2.4 to be a mere low-rank perturbation of the input at any depth, seems overly simplistic of the way that real networks shape and transform input distributions.
-  Training: While the paper makes brief remarks on training, given that one of the key findings of the paper is on NTK, more exploration and discussion of training dynamics (though it is acknowledged as outside scope) would probably add some value

**Questions For Authors:**

- My most important lingering question is, most classical results in free probability either work  for independent features, or if they are dependent but follow Gaussian distributuion , so their dependence is captured by the covariance. However, once we pass them through one layer of non-linear activations, neither of which is true. How can we get the eigenvalue distribution of the conjugate and tangent kernel here? Is the key the class-structure in the inputs and the resulting low-rank perturbation that enables the rest of the analysis?
- The follow-up to the previous question is could your finite-rank spike analysis be extended to more general distributions than the GMM? For example,  what if instead of class-structure, we have an input whose eigenspectrum of covariance is explicitly given to us? Regardless of the spiked eigenvalues structure, Can a similar analysis reveal the bulk eigenspectrum conjugate and tangent kernel ?

**Relation To Broader Scientific Literature:**

The paper extends and complements a line of research studying  neural kernels in high dimensions using random matrix theory.

- CK/NTK in Infinite Width: Prior works (Jacot et al. 2018;) derived closed-form expressions for infinite-width kernels but did not analyze a mini-batch of samples, and hence analyze spiked eigenvalues beyond the main spectrum. The present paper looks at how mixture-structure in data can cause outliers in large finite dimension.
- Spiked Models in RMT: The approach of analyzing finite-rank deformations of a baseline limiting distribution is reminiscent of Baik et al. (2005) and other spiked covariance matrix models. Here, the baseline is the limiting kernel, and the finite-rank perturbation arises from class-structured data.
- Empirical CK/NTK Spectra: Related to works by Fan & Wang (2020) or Wang et al. (2024), who studied the spectral distribution in large neural networks. This paper  approach is similar but focuses specifically on outliers and their eigenvectors in presence of cluster / class structure.
- Feature Learning & Data Clusters: The demonstration that the outlier eigenvectors align with class membership echoes analyses in
Worth improving:
- “Neural Collapse”: I thin the connection of this result to neural collapse worth discussing. While the paper is about a random state / initialization, one could view this class-structure as a form of neural collapse or other clustering phenomena in wide networks, but from a pure RMT perspective. This might be interesting for some readers.
- "Feature learning": because we know that classical NTK  is not leading to learning dynamics, it might be worth making some statements of how these results could affect feature learning in middle layers? In that sense, some brief mention and contrast with feature learning literature (Tensor Programs, muP literature) might also be worthwhile.

**Theoretical Claims:**

- The main theorems (Theorem 3.1 & 3.2) rely on finite-rank perturbation arguments from random matrix theory. The approach of showing that the empirical kernel matrix is well-approximated by a low-rank correction to a known limiting distribution is standard.
- The authors reference prior results on deterministic equivalents for CK and NTK from (Gu et al., 2024) and from classical RMT references (e.g., Bai & Silverstein). The extension to identifying spiked eigenvalues and projecting onto the eigenspace (Theorem 3.4) is also sound
- While I did not check all the proofs in depth, I did not spot an obvious algebraic or conceptual flaw, they seem sensible in the sense that they rely on well known established lemmas (like Woodbury identity, resolvent expansions, Schur complement formula, etc.).  However the proofs are quite technical and dense, and I might have missed something.
- One area that I am less certain about:  In a multi-layer setting, different features are tied/coupled together, and they are not necessarily a Gaussian matrix to approximate their eignevalues. Can authors explain how they can seeming go around this dependencies to be able to apply free probability on deep representations? What are the key facts/lemmas that enable this?

---

> ### Author Rebuttal · Authors · 2025-03-31
>
> We sincerely thank the reviewer for his/her encouraging feedback. The recognition of our work's novelty, technical rigor, and clarity is highly motivating. We also greatly appreciate the insightful comments provided. In the following, we provide point-by-point responses (marked with $\Huge{\cdot}$). The statements of comments have been condensed due to character limitations.
> > The paper already cites much of the very relevant literature. But it ... two lines of theory.
> * We thank the reviewer for sharing these references, which will be incorporated into the revised version.
>
> > Data Model & Assumptions: ... distributions.
> * GMMs are widely used in machine learning analysis due to the technical convenience. In large-scale neural network models, mixture model behave asymptotically as if GMMs (Seddik el al. (2020, ICML); Couillet and Liao (2022, Cambridge University Press)). In fact, from the perspective of random matrix theory, many properties of Gaussian random matrices can be extended to random matrices with general distributions under certain moment conditions—a phenomenon known as universality (Ding and Yang (2018, AOAP)). To support our findings, we perform an experiment on the MNIST dataset. The visualization of the spectrum and eigenvectors of the CK matrix can be accessed via the following link:   <https://anonymous.4open.science/r/eig5492/README.md>. The spectrum reveals four isolated eigenvalues. Notably, the eigenvector corresponding to the largest eigenvalue contains group features, whereas the remaining three appear to lack such information. These empirical observations are consistent with our theoretical results presented in Theorem 3.4 through Remark 3.6.
>
> > Training: While ... some value.
> * We appreciate the suggestion to discuss training dynamics, and while we agree it would be valuable, space constraints prevent us from including it here. Instead, we kindly refer you to our detailed response to Reviewer Hb59’s second question, where this topic is addressed in detail.
>
> > As mentioned earlier, some discussion around neural collapse ... readers.
> * We acknowledge the importance of discussing the neural collapse phenomenon in relation to our study, as both lines of work consider input data with inherent cluster structures. Neural collapse illustrates that when training error reaches zero, classifiers tend to converge towards class means. In contrast, our research demonstrates that even without training, certain eigenvectors of kernel matrices (NTK and CK) derived from neural networks can encode group features within isolated eigenvalues. We will provide discussions on neural collapse and feature learning in the revised version.
>
> > The paper celebrates its key finding ... takeaways.
> * According to Lemma 2.4, the bulk eigenvalue distributions of the CK matrix exhibit similar curves across different activations, differing only by a constant scaling factor. However, Figure 4(b) shows an isolated eigenvalue for the ReLU activation, whereas Figure 4(a) does not display this phenomenon for the polynomial activation. This observation aligns with our findings on the influence of activations on the emergence of isolated eigenvalues. In the revised version, Section 3.3 will clearly elaborate on this point to highlight the effect of different activations.
>
> > My most important lingering question ... analysis?
> * Firstly, we would like to clarify that our analysis is based on the asymptotic spectral equivalence of kernel matrices, as established by Lemmas 2.4 and 2.5 in our paper. This equivalence allows us to operate within a low-rank perturbation framework, thereby circumventing the technical complexities arising from  dependencies introduced by nonlinear activation functions. Consequently, our results are derived using standard random matrix theory tools, without the need for free probability theory techniques.
>
> > The follow-up to the previous question ... kernel?
> * Our theoretical results hold for the GMM model. If the eigenspectrum of the covariance matrix is explicitly given without any class structure, it simplifies to a special case of GMM with only one group. Consequently, the asymptotic spectral equivalence holds for the CK and NTK matrices, as stated in Lemmas 2.4 and 2.5. This reduction allows us to focus on investigating the isolated eigenvalue and eigenvector of matrices of the form: $$X^TX+a\mathcal{1}\mathcal{1}^T+b\psi\psi^T+c\mathbf{I},$$ where $\mathcal{1}$ and $\psi$ are two vectors. Under certain conditions on $(a\mathcal{1}\mathcal{1}^T+b\psi\psi^T)$, isolated eigenvalues may emerge. Regarding bulk eigenvalues and eigenvectors, the limiting spectral distribution is provided in Lemma 2.6. Further theoretical analysis, such as the rigidity of the bulk eigenvalues and the delocalization of eigenvectors, are beyond the scope of this paper. However, we believe that it is feasible to establish the local behavior of the bulk spectrum if the local laws for CK and NTK matrices can be derived.

---

### Official Review · Reviewer_Hb59 · 2025-03-26

**Overall Recommendation:** 1

**Summary:**

This paper studies outlier eigenvalues and eigenvectors of conjugate and neural tangent kernels for multi-layer fully connected neural networks at random initialization. The dataset can be a general Gaussian mixture model. This result shows how the information of the group features in the dataset propagates through the multiple layers via the outlier eigenvalues and corresponding eigenvectors.

**Claims And Evidence:**

Yes, the claims made in the submission are supported by clear and convincing evidence.

**Essential References Not Discussed:**

There are more references that should be mentioned or discussed in the paper:

1. Ba et al 2023 (https://openreview.net/forum?id=HlIAoCHDWW) studied the Gaussian data with one spiked direction in the population covariance matrix and used the spike in conjugate kernel to study the feature learning.

2. There are several papers that considered a similar Gaussian mixture setting and random neural networks: Liao, Couillet 2018 https://proceedings.mlr.press/v80/liao18a/liao18a.pdf, Couillet, Liao, Mai 2018 https://ieeexplore.ieee.org/document/8553034, Liao, Couillet 2019 https://ieeexplore.ieee.org/document/9022455, Ali, Liao, Couillet 2020 https://openreview.net/forum?id=qwULHx9zld. It would be better to have some discussions on these papers.

3. Liao, Couillet, and Mahoney 2020 (https://papers.nips.cc/paper/2020/hash/a03fa30821986dff10fc66647c84c9c3-Abstract.html) studied the random Fourier feature model and the random feature regression model in this case.

**Experimental Designs Or Analyses:**

Yes, the numerical experiments of the paper validate the results of its theoretical analysis.

**Methods And Evaluation Criteria:**

No, this paper doesn't propose any method or evaluation criteria.

**Other Comments Or Suggestions:**

See weaknesses.

**Other Strengths And Weaknesses:**

## Strengths:
This presentation is clear and well-organized. The mathematical results are clearly presented.

## Weaknesses:
1. Wang et al., 2024 in the reference have studied the propagation of the spike eigenvalues and eigenvectors for conjugate kernel matrix through multiple layers. I do not know the novelty of this paper comparing with Wang et al., 2024. Wang et al., 2024 also have similar simulation for gaussian mixture dataset. There should be a clear comparison with this paper. Do the results of Wang et al., 2024 totally cover the results of CK in the current paper?

2. Although the mathematical results presented in Section 3 are clearly presented, there is a lack of explanations regarding the statistical implications of these results. Can we derive any practical applications or insights for the machine learning community from these findings? Even some case studies or remarks should be better included in the main text.

3. A comprehensive literature review should be included to provide context and establish the relevance of the work. Additionally, a discussion should be presented outlining the limitations of the current study.

**Questions For Authors:**

See weaknesses.

**Relation To Broader Scientific Literature:**

This paper uses random matrix theory to understand the extreme eigenvalues and eigenvectors of empirical kernels in random neural networks.

**Theoretical Claims:**

Yes, I checked the correctness of the proofs for theoretical claims. The results are basically based on previous works, e.g.Benaych Georges & Couillet, 2016.

---

> ### Author Rebuttal · Authors · 2025-04-01
>
> We sincerely thank the reviewer for acknowledging our paper as well-organized and clearly presented, and we appreciate the constructive comments. Below, we provide point-by-point responses (marked with $\Huge{\cdot}$). The statements of weaknesses have been condensed to comply with character limitations.
>
> > Wang et al., 2024 in the reference ... paper?
> * We thank the reviewer for this important comment. The pioneering work by Wang et al.(2024) offers elegant results on the spiked eigenvalues and eigenvectors of CK. To clarify the differences between Wang et al.(2024) and our study, we provide the following comparisons.
> 1. We allow for different population covariance matrices $C_a$ in GMM, without relying on the $\tau_n-$orthonormal assumption introduced in Wang et al. (2024). Specifically, this assumption requires the difference between $tr(C_a)$’s to be $o(p^{2/3})$, whereas our study only requires this difference to be $O(p)$.
> 2. We do not require $X^TX$ to contain isolated eigenvalues, as specified in Assumption 2 of Wang et al.(2024). Instead, the isolated eigenvalues of CK and NTK can arise from the group features of input data, i.e., through the terms $VA_{\ell}V^T$ and $VB_{\ell}V^T$ in our equations (5) and (8).
> 3. They assume isolated eigenvalues are simple, focusing on single eigenvectors in their main theorems. In contrast, our work accommodates isolated eigenvalues with multiplicities greater than one and analyzes the corresponding eigenspaces.
> 4. The two studies adopt different assumptions regarding the activation functions. Their work employs weaker conditions on the moments of the derivatives, whereas ours imposes weaker constraints on the expectations of the first and second derivatives.
> 5. Given NTK's ability to approximate the generalization and dynamics of neural networks (NNs) on real data, our work explores the properties of NTK which are not addressed in Wang et al.(2024).
>
> Additionally, we would like to point out a typo: the condition $trC_a^o=O(\sqrt{p})$ in Assumption 2.1 is unnecessary. This correction, along with the aforementioned comparisons, will be included in the revised version.
>
> > Although the mathematical ... text?
> * For NTK: Let $\boldsymbol{y}$ represent the label vector of the data. When the loss function $L(t)=\|\|\boldsymbol{y}-\hat{\boldsymbol{y}}(t) \|\|^2 $, where $\hat{\boldsymbol{y}}(t)$ denotes the prediction at time $t$, the time evolution of the residual is approximately given by the following ODE: $$\frac{d}{dt} \hat{\boldsymbol{y}} (t)= \boldsymbol{K}_{\text{NTK}} \big(\boldsymbol{y} -\hat{\boldsymbol{y}} (t)).$$ This indicates that NTK captures the dynamics of the training process in ultra-wide NNs. We draw the following statistical implications:
> 1. Based on our theoretical results from Theorem 3.7 to Remark 3.9, the first-order limits of entries in the isolated eigenvectors may or may not contain group features.
> 2. When the eigenspace associated with the largest isolated eigenvalue contains group features, NNs tend to prioritize learning this information. Conversely, if this eigenspace lacks group features, NNs instead prioritize learning irrelevant information, diverting attention away from effective group features.
> 3. For non-isolated eigenspaces, we conjecture that their first-order limits contain significantly fewer group features, as observed in classical random matrices like the sample covariance matrix, where the entries of non-spiked eigenvectors are delocalized under mild conditions. Consequently, after effectively learning the group features captured by eigenspaces corresponding to isolated eigenvalues, NNs primarily shift their focus to learning from eigenspaces that lack first-order group features. In GMM, this implies that NNs begin leveraging more noise to learn $\boldsymbol{y}$, leading to overfitting.
>
> This potentially provides a perspective based on NTK to understand the phenomenon of overfitting. We will include this discussion in the revised paper.
> * For CK: We conducted an experiment on the MNIST data. A histogram showing the spectrum of the third layer's CK matrix and line plots of the eigenvectors for the top 5 eigenvalues can be found at the following link: <https://anonymous.4open.science/r/eig5492/README.md>.
> The spectrum shows four isolated eigenvalues, with the eigenvector associated with the largest eigenvalue being informative, while the other three appear non-informative. These observations align with our theoretical findings from Theorem 3.4 to Remark 3.6.
>
> > A comprehensive literature ... study.
> * We will add all works mentioned by the reviewer, as well as additional relevant studies, including Yang and Salman (2019,arxiv), Bai and Lee(2020,ICLR), Hron et al.(2020,ICML), Papyan et al.(2020,PNAS), Yang(2020,arxiv), Wang et al.(2023,NIPS), Dandi et al.(2024,arxiv), and Engel et al.(2024,ICLR). Moreover, we will discuss potential extensions such as relaxing the distribution assumption and considering other architectures.

---

### Decision · Program_Chairs · 2025-05-01

**Decision:**

Accept (poster)

**Comment:**

This paper studies the outlier eigenvalues and eigenvectors of conjugate and neural tangent kernels for fully connected deep neural networks having random weights, with input data drawn from a general Gaussian mixture model.

The authors have done a commendable job during the rebuttal, and the majority of reviewers are now **convinced of the significance and merit of the work**.

As noted by Reviewer Hb59 and others, there remains room for improvement, particularly in terms of:
1. Clarifying the differences between this paper and Wang et al., 2024; and
2. Emphasizing the practical implications of the proposed theoretical analysis for ML applications.

Nevertheless, given the importance of the problem and the rigorous and technically solid contributions, I believe this paper will be a valuable addition to the ICML community and a useful reference for future work.

As a result, I recommend acceptance of this paper to ICML 2025.
Please ensure that the additional numerical results, clarifications, and discussions provided during the rebuttal phase are included in the final version.